# Influence of Muscle Type on Physicochemical Parameters, Lipolysis, Proteolysis, and Volatile Compounds throughout the Processing of Smoked Dry-Cured Ham

**DOI:** 10.3390/foods10061228

**Published:** 2021-05-28

**Authors:** Nives Marušić Radovčić, Ivna Poljanec, Sandra Petričević, Leticia Mora, Helga Medić

**Affiliations:** 1Faculty of Food Technology and Biotechnology, University of Zagreb, Pierottijeva 6, 10000 Zagreb, Croatia; ipoljanec@pbf.hr (I.P.); hmedic@pbf.hr (H.M.); 2Croatian Veterinary Institute, Regional Institute Split, Poljanička Cesta 33, 21000 Split, Croatia; petricevic.vzs@veinst.hr; 3Instituto de Agroquímica y Tecnología de Alimentos, CSIC, Avenida Agustin Escardino 7, 46980 Paterna, Valencia, Spain; lemoso@iata.csic.es

**Keywords:** dry-cured ham, smoking, processing, aromatic profile, GC-MS, physicochemical parameters, lipolysis, proteolysis

## Abstract

The influence of muscle type (biceps femoris, BF and semimembranosus, SM) on physicochemical parameters, volatile compounds, and the extent of proteolysis and lipolysis during the manufacturing of smoked dry-cured ham was investigated. A total of fifty smoked hams were sampled: raw ham, after salting, smoking, drying, and ripening. Almost all physicochemical parameters were affected by muscle type, manufacturing stage and their interactions. SM had lower water, ash, NaCl content, and water activity (a_w_), while fat and protein content were higher after ripening compared to BF. BF showed higher L*a*b* values compared to SM. The results of texture profile analysis showed that almost all analyzed parameters were influenced by muscle type and production stage. A total of 88 volatile compounds were identified, showing an increase in its number during processing: 31 volatile compounds were identified in raw ham and 72 after the ripening phase. Aldehydes and phenols were the predominant groups of compounds, followed by alcohols, ketones, aromatic hydrocarbons, aliphatic hydrocarbons, esters, and terpenes. Muscle type and production phase significantly affected lipid oxidation and the index of proteolysis: in SM, thiobarbituric acid reactive substances (TBARS) increased faster than in BF, while proteolysis had an opposite effect and was more pronounced in BF.

## 1. Introduction

Dry-cured ham is a meat product of excellent quality and it is produced mainly in the Mediterranean region. The largest producers are Italy, Spain, and France. The Croatian Dalmatian dry-cured ham (Dalmatinski pršut) is a traditional meat product with protected geographical indication (PGI) and is produced in the southern part of Dalmatia, Croatia. It differs from other types of dry-cured ham because one of the stages of production involves smoking that adds desirable sensory characteristics to dry-cured meat products. Phenolic components of the smoke contribute significantly to the aroma of dry-cured meat products. The smoke acts like a barrier against the development of rancidity with its protective film on the surface of the smoked dry-cured ham [1]. Dalmatinski pršut is produced according to the traditional processing method with sea-salt and without any other additional additives [1]. During dry-curing, many complex biochemical changes occur, including the degradation of proteins and lipids. However, physicochemical, aromatic, textural, and sensory properties of dry-cured ham vary considerably depending on the changes in the technological production [2] as well as on the influence of the breed of pig used in production (rearing system, feed, genetic type, etc.) [3].

Physicochemical changes throughout the dry cured ham processing have been studied for different types of dry-cured ham: Spanish Iberian [4,5,6], Serrano [7] and Celta [8], French Bayonne [9], Italian Toscano [10,11], Chinese Jinhua ham [12], Polish Kumpiak podlaski [13]. These studies concluded that the production process has a significant impact on physicochemical parameters, with varying intensity depending on muscle types [8]. The two most studied muscles in dry-cured ham, the biceps femoris (BF) and the semimembranosus (SM), are exposed to different conditions during dry-cured ham processing and show a different time chart of proteolysis [9]. The SM is an external muscle that is subjected to a faster dehydration and salt intake during the first stages of processing. The internal muscle, BF, shows higher water content and proteolytic activity during the production process than SM [14]. These differences between muscle types leads to a different intensity of proteolysis and lipolysis, and thus to a different number of volatile compounds in both BF and SM [15].

The flavor of dry-cured ham is one of the most important quality attributes that influence consumers’ acceptance. The two main processes that contribute to the final quality of dry-cured ham are proteolysis and lipolysis [16,17]. During the ripening phase, degradation of proteins and lipids to amino and fatty acids occurs due to the action of endogenous enzymes. The products of this degradation are the main contributors to the aroma of dry-cured ham [2]. In recent decades, volatile compounds analysis has been frequently performed to characterize dry-cured meat products, especially dry-cured ham. Furthermore, the differentiation of dry-cured ham based on the production process as well as different muscles represents an interesting topic in the field of meat science. Although the influence of different processing techniques has been extensively studied for other types of dry-cured ham, particularly Iberian, Toscano, San Daniele, and Parma ham [10], no data was discovered on the formation of volatile compounds as well as other important physicochemical parameters in different muscles for smoked dry-cured ham. Therefore, the aim of this study was to evaluate the influence of muscle type (BF or SM) on volatile compounds during the production of smoked dry-cured ham Dalmatinski pršut. In addition, the evaluation was carried out based on the physicochemical parameters (water, fat, protein, ash content, water activity, NaCl concentration, and pH) as well as instrumental texture and the extent of proteolysis and lipolysis.

## 2. Materials and Methods

### 2.1. Dry-Cured Ham Samples

A total of 50 hams were used in this study. The raw hams weighing 11.90 ± 0.50 kg were obtained from Duroc × (Yorkshire × Landrace) breed pigs. The dry-cured hams were produced following the PGI specification for the production of smoked dry-cured ham, Dalmatinski pršut, until they reached the final weight of 7.50 ± 0.40 kg. The dry-cured hams were produced as follows: salting up to 30 d (0–5 °C, RH from 80%–90%), cold smoking 45 d (<22 °C), drying up to 3 months (12–16 °C; RH reduced from 90 to 70%) and ripening 6 months (12–15 °C, RH 65%–75%). Hams were dry-salted with excess of coarse sea salt. A heap was formed by alternating layers of ham samples and layers of salt. In this way, the samples were totally covered with salt. Smoking of whole ham pieces was done using cold smoke obtained by burning hardwood (*Fagus* sp.) and Oak (*Quercus* sp.). The temperature in the smoking chamber did not exceed 22 ° C. Total time of production was 12 months. Regarding PGI specification, at the end of ripening the smoked dry-cured ham, Dalmatinski pršut, should have water activity below 0.93 and water content between 40%–60% and after 12 months of production, the dry-cured ham is ripe and ready for consumption.

The BF and SM muscles were sampled from ten randomly selected hams at the following stages of production: raw ham (day 1), after salting (26 d), after smoking (45 d), after drying (75 d) and after ripening (12 months of processing). Samples were coded, vacuum packed, frozen, and stored at −18 °C. Prior to analysis, samples were thawed for 24 h at 4 °C. All analyses were performed in triplicate.

### 2.2. Physicochemical Analysis

Moisture, fat, ash content, and NaCl were determined according to official methods [18]. Fat content analysis was performed following the AOAC method [19]. Protein content was determined according to the Kjeldahl method [20]. Non-protein nitrogen was determined as described by [21]. The proteolysis index was calculated from the ratio between the non-protein nitrogen and the total nitrogen determined by the Kjeldahl method [20]. Water activity (a_w_) was measured using a LabMaster a_w_-meter (Novasina, Switzerland) according to the manufacturer’s instructions. For pH measurement, a suspension of 10 g of sample of smoked dry-cured ham was mixed with 90 mL distilled water, and pH was measured by pH-meter (benchtop sensION tm + MM374, Hach Company, Loveland, CO, USA). Thiobarbituric acid reactive substances (TBARS) were quantified to determine the degree of lipid oxidation according to the method described by [22] with slight modifications. Muscle samples (5 g) were dispersed in 25 mL of 5% TCA and then treated with 10 mg of BHT to prevent further oxidation. The mixture was homogenized on ice for 60 s (Ultra Turrax T18 basic, IKA Werke GmbH & Co. KG, Baden-Württenberg, Germany) and centrifuged (10 min, 12,000 rpm, 4 °C) (Rotina 380 R, Hettich LabTechnology, Tuttlingen, Germany). The supernatant was filtered through filter paper (Whatman N° 54, Whatman, Dassel, Germany). An aliquot of 4 mL was treated with 4 mL of 0.02 M TBA solution. The mixture was heated for 1 h at 100 °C in a block heater (Stuart SBH130D, Cole-Parmer Ltd., Stone, UK). The absorbance was measured at 532 nm (Specord 50 Plus, AnalytikJena, Jena, Germany) using 1,1,3,3-tetra-methoxy-propane (TMP) as standard. The results were expressed as mg malonaldehyde (MDA)/kg sample.

### 2.3. Colour

CIE surface color parameters (L*: lightness, a*: redness, b*: yellowness) (CIE, 1976) were measured using a Minolta CM-700d spectrophotometer (Minolta, Japan) with illuminant D 65 10° standard observer, 8 mm aperture, with open cone. Each sample of BF and SM muscle was analyzed in six replicates, avoiding regions of fat to measure lean muscle color.

### 2.4. Texture Profile Analysis

A texture profile analysis (TPA) of the dry-cured ham samples was carried out using a texture analyzer (Ametek Lloyd Instruments Ltd., West Sussex, UK). The texturizer was equipped with a 50-kg load cell supported by NexygenPlus software. The analysis was performed at room temperature. Prior to analysis, the samples were cut into 10 × 10 × 10 mm and conditioned for 2 h at 20 °C. The sample was compressed twice at a crosshead speed of 1 mm/s to 50% deformation (the rest period between cycles was 5 s), and the following parameters were determined from the force-distance curves: Hardness (N), gumminess (N), chewiness (N × mm), adhesiveness (N × mm), springiness (mm), cohesiveness, and resilience.

### 2.5. Analysis of Volatile Compounds

The analysis of volatile profiles was performed by solid phase microextraction (SPME) and their qualification and quantification on GC/MS following the method by Marušić et al. (2011) [23]. For sample preparation, 5 g of minced BF or SM muscle was mixed with 25 mL of distilled water saturated with NaCl in a commercial blender. A 10 mL aliquot was added to 20 mL vials with the addition of 100 μL 4-methyl-2-pentanol (1.2 mg/kg) (internal standard) and magnetic stirrer, and then sealed with a PTFE septum. An SPME fiber coated with 2 cm of 50/30 μm DVB/Carboxen/PDMS (Supelco, Bellefonte, PA, USA) was placed over the sample mixture (previously heated at 240 °C for 2 min). Extraction of triplicate 20-mL vials was performed at 40 °C for 180 min with constant stirring. After extraction, the SPME fiber was transferred to the injection port of the 6890N gas chromatograph coupled to a 5975i mass selective detector (Agilent Technologies, Santa Clara, CA, USA). Helium was used as the carrier gas with a constant flow of 1 mL/min (9.59 psi). The column used for volatile separation was a DB-5MS capillary column (30 m × 0.25 mm, film thickness 0.25 μm (Agilent Technologies, Santa Clara, CA, USA)). Injection was performed in splitless mode, with an injector temperature of 230 °C and a desorption time of 5 min. The chromatographic conditions were: The temperature was set at 40 °C, isothermal for 10 min, then increased to 200 °C at a rate of 5 °C/min, and then to 250 °C at a rate of 20 °C/min. The final temperature was held for 5 min and the transfer line temperature was maintained at 280 °C. Mass spectra were recorded at 70 eV at a rate of 1 scan/s over the *m*/*z* range of 50–450.

To calculate the retention indices (RI) of the detected compounds, an in-house mixture of C8-C20 n-alkanes was run under the same chromatographic conditions. The program AMDIS 3.2 version 2.62 was used to identify the components using the spectral library NIST 2005 version 2.0 (NIST, Gaithersburg, MD, USA) and to compare the retention indices obtained with literature values ([24] and the in-house library).

### 2.6. Statistical Data Analyses

One-way analysis of variance (ANOVA) was used for BF and SM for physicochemical and volatile compound data. One-way ANOVA was followed by a post hoc Tukey honest significance (HSD) if a significant effect was found (*p* < 0.05). A Student *t*-test (*p* < 0.05) was performed to test for differences in physicochemical and volatile compound data between investigated muscles at the same processing stage. The general linear model (GLM) was used to estimate the effects of the factors (muscle (M); phase of production (P) and M × P) on the studied parameters (dependent variables). GLM test was performed for a significance level *p* < 0.05. All statistics were generated using the statistical program SPSS v.9.0 (SPSS inc., Chicago, IL, USA).

## 3. Results and Discussion

### 3.1. Changes in Physicochemical Parameters during the Production of Smoked Dry-Cured Ham

During dry-cured ham production, intense biochemical reactions take place, resulting in significant changes in physicochemical composition. These changes are the result of proteolytic and lipolytic activities [25]. In (Table 1), the physicochemical parameters in two muscles (BF and SM) during the production process of smoked dry-cured ham, Dalmatinski pršut, are presented. As can be seen from the obtained results, the chemical composition of the muscle types BF and SM was significantly affected during different stages of the production process. The water content decreased in both BF and SM muscles from 74.82 and 74.42% in the raw ham to 56.78 and 42.54% in the final product, respectively. The decrease in water content was slower in the BF muscle than in the SM due to the different position in the ham: BF is covered with skin and subcutaneous fat and therefore water evaporates more slowly in contrast to the SM muscle. There were significant differences (*p* < 0.05) between the muscles at all stages of the production, except for the raw ham. Dehydration was more intense during the ripening phase, due to the duration of this phase as well as the environmental conditions (higher temperature and lower relative humidity). As shown in (Table 4), both muscle (M) and the production phase (P) and their interaction (M × P) significantly (*p* < 0.05) affected the water content during the processing of smoked dry-cured ham.

Intramuscular fat (IMF) was influenced by M, P, and their interactions (M × P) (Table 4). It significantly decreased from an initial value of 5.54 and 6.00% to 2.57 and 5.55% of dry matter (DM) after the smoking phase for BF and SM muscles, respectively. IMF content significantly increased after the drying and ripening phase, resulting in a higher content in SM (14.86% of DM) than in BF (12.45% of DM) at the end of the process. The higher IMF content that was found in SM over BF was also found in other types of dry-cured ham [14]. Higher IMF content in the final product was also found in Jinhua ham [26]. The fat content in this study was higher than those reported for Bayonne (3.5%) and Corsican (5.3%) hams [27]. Different fat contents are a consequence of the different breeds of pigs and the different feeding systems [28]. Moreover, IMF content is the parameter that most influences the appearance, texture, and flavor of dry-cured ham [29].

A significant (*p* < 0.05) decrease in protein content was observed in both muscle types during the production process: from an initial value of 88.19 and 89.54% of DM in raw ham to 69.16 and 74.72% of DM at the end of the process in BF and SM, respectively. Similar results were also observed in other types of dry-cured ham such as Celta [8]. This decrease in protein content could be the result of the drying process, leading to a decrease in the percentage of protein in the total solids content of the samples. Furthermore, decrease of the protein content probably occurred due to the absorption of NaCl after salting and subsequent diffusion in muscles during processing, resulting in a lower NaCl content in BF muscle in the initial stages of production and a higher content at the end of the process [8]. A significant (*p* < 0.05) decrease in protein content was observed in both muscles after the salting stage, and it was more pronounced in SM muscle due to its external position and greater exposure to environmental conditions. Compared to studies on other types of dry-cured ham, the protein content in BF is lower than in SM muscle [8]. A decrease in protein content is also observed after the smoking phase, but this is due to NaCl diffusion in the BF, as cold smoking has no effect on protein content in dry-cured hams. A study supporting this by Martuscelli et al. (2009) [30] concluded that different smoking methods had no effect on protein content in BF of dry-cured ham.

NaCl content was significantly (*p* < 0.05) affected by muscle type, the production phase, and their interaction (Table 4), and increased significantly during the salting, smoking, and drying phases of production. This is a consequence of the salt distribution throughout the ham. A different trend was observed depending on the muscle type: in the BF muscle, the highest increase was observed after the smoking phase (from 0.20 in raw ham to 12.47% of DM), after which it increased slowly until the end of the process (to 17.55% of DM). In contrast, the salt content in the SM muscle increased rapidly after the salting phase (from 0.18 to 16.78% of DM), as this muscle is in direct contact with salt during the salting phase and remained constant after the drying phase. At the end of the production process, a significant decrease was observed (11.20% of DM). A slower increase was observed in the muscle BF compared to the muscle SM because in BF the salt is homogeneously distributed throughout the ham muscle during the production process. The NaCl content after the ripening phase was higher (*p* < 0.05) in the BF than in the SM which is in agreement with its higher water content. This is also confirmed by other authors [31]. The SM muscle is exposed to a higher concentration of salt during salting and to a higher dehydration during drying and has a higher salt content than the inner BF muscle after the salting, smoking, and drying phases. However, it should be noted that throughout the production process there is a tendency to equalize the salt concentration throughout the piece, therefore the salt equalization process results in a higher salt content in BF compared to SM after the ripening phase. The diffusion of salt into the muscle tissue from the SM muscle with a higher salt content continues due to the existence of a concentration gradient, which ultimately leads to the inner BF muscle with a higher water content also having a higher salt concentration [31], as found in this study. The increase in ash content in both muscles followed the same trend as NaCl as ash content depends on NaCl.

Water activity (a_w_) was significantly (*p* < 0.05) affected by M, P and M × P interactions (Table 4). a_w_ gradually decreased over the production period from an initial value of 0.97 and 0.96 to 0.88 to 0.86 in BF and SM, respectively. This decrease was more pronounced in the SM muscle and is due to the decrease in water content, salt diffusion, and the intense dehydration experienced by the ham during the dry-ripening phase [29].

The pH was only affected by production (Table 4). An increase in the final pH was observed compared to the pH of the raw ham (from 5.83 to 5.93 and 5.78 to 5.99 in BF and SM, respectively). This increase in pH due to the production process is also reported in other types of dry-cured hams and could be related to low-weight nitrogen molecules and ammonia formation due to proteolytic enzyme activity [29].

### 3.2. Instrumental Measurement of Color and Texture of Smoked Dry-Cured Ham

The color of dry-cured ham is one of the most important appearance characteristics and influences of consumer choices when purchasing sliced dry-cured ham [32]. Table 2 shows the color parameters of smoked dry-cured ham during the production process (L*a*b*) of BF and SM. The L*, a*, and b* values were significantly affected by both the muscle and the production stage, while their interaction affected the L* and a* values (Table 4). The processing effect was more pronounced in the SM muscle. BF and SM muscles are submitted to different conditions during the dry-cured ham process. BF is covered with the skin and thick layer of fat (internal muscle) and SM muscle is superficial with no fat cover (external muscle) [8]. Therefore, SM is more exposed to environmental conditions due to its external position and thus affects the color parameters, especially the L* value. Lightness (L*) depends on the moisture movement (dehydration) towards the surface [32] and decreases throughout the production process. This decrease is directly related to the water content. Other authors found that water loss increases the concentration of pigments and leads to a decrease in L* value. It is also found that L* values decrease with increasing salt concentration [32] which is in agreement with the results of this study and other studies for Istrian and Iberian dry-cured ham [23,33].

The a* values were significantly higher (*p* < 0.05) for the BF muscle than for the SM in the raw ham and in the ham after the drying and ripening phase, showing that BF had a more intense color than the SM muscle. The a* value was associated with intense drying of dry-cured ham [4], so this may be the reason why higher a* values were found in BF, due to significant differences in water content between the two muscles. Finally, b* values decreased with increasing processing time, and the decrease was more pronounced after the smoking stage (Table 2), where the differences between muscles were significant, and this trend continued until the end of the production process. This is also comparable to the results obtained in other types of dry-cured ham [32,34]. Changes in color parameters were also observed in other types of dry-cured hams such as Celta ham, where BF had higher L*, a*, and b* values than the SM muscle in the last phase of production [8]. Other authors found the same trend in Teruel dry-cured ham [34]. The L*a*b* values of BF and SM determined in this study showed the same trend as in Iberian dry-cured hams [4].

The texture of dry-cured ham has a major impact on the product quality perception of consumers and is determined by proteolytic degradation that occurs during dry-curing. The relationship between proteolysis and texture during the process of dry-curing is well established [9], and salting and ripening have been reported as two stages with the greatest influence on proteolysis and consequently meat texture [35].

Table 3 shows the changes in texture profile analysis (TPA) parameters of BF and SM of smoked dry-cured ham during the production process. Cohesiveness was the only TPA parameter not significantly (*p* > 0.05) affected by muscle and all analyzed TPA parameters were significantly (*p* < 0.05) affected by the production phase. Their interaction during the production process significantly (*p* < 0.05) affected all analyzed parameters except springiness (Table 4).

Hardness (N) increased significantly in both muscles during the initial production phases. An increase in hardness was more evident in SM, especially after salting and drying, which can be attributed to the increasing salt concentrations and water loss (Table 1). Several studies showed a strong relationship between hardness and water and salt content [9,31,36]. The sharp decrease in hardness occurred after the ripening phase in both muscles, probably due to the progression of proteolysis (Figure 1B). Similar to our results, another study [21] also reported a decrease in the hardness of BF and SM muscles in later stages of production in Bayonne dry-cured ham. This phenomenon is related to the initial loss of protein solubility which leads to hardening of muscles but is followed by degradation of myofibrillar proteins which cause the tenderization of muscles in later phases of production [21]. Although proteolysis was more advanced in BF between the drying and ripening phase (Figure 1B), this muscle did not suffer a greater decrease in hardness values than SM. This effect is likely a consequence of the incorporation of the salt and the additional water loss in BF (Table 1), which probably resulted in higher hardness values that would otherwise decrease due to more intense proteolysis in BF muscle. The final hardness values were significantly higher in SM than in BF, which is consistent with the results of other studies on different types of dry-cured ham [14,34,37].

Changes in gumminess and chewiness are also related to water content [36,38] and followed the same trend as hardness in investigated muscles. BF showed significantly lower values of gumminess and chewiness at the end of processing, in agreement with the results observed after 12 months of processing of Kraški pršut [14].

Adhesiveness is a parameter strongly influenced by proteolytic processes [35]. Adhesiveness progressively decreased during processing in both muscles. A decrease in adhesiveness values in BF and SM during 12 months of Bayonne ham processing was also reported [9], but with much lower values than those found in this study. Decrease in adhesiveness during dry-cured ham production is associated with drying and protein gelatinization, resulting in improved sliceability of the final product [39]. We can attribute the decrease in adhesiveness in both muscles during processing of the smoked dry-cured ham to drying and progressive loss of water content (Table 1). Significantly higher (*p* < 0.05) values for adhesiveness in BF compared to SM throughout processing may be related to less severe drying due to the internal position of BF in the ham and, presumably, more advanced proteolysis at the end of processing. Final adhesiveness values were significantly higher for BF over SM, which was also reported for Bayonne [9] and Teruel [34] dry-cured hams after 12 months of processing.

Springiness and cohesiveness are TPA parameters negatively correlated with a_w_ and water content [31]. Springiness and cohesiveness decreased significantly over time in both muscles, with higher final values in BF. The lower springiness and cohesiveness in SM may be attributed to the more intense dehydration in this muscle, based on previous studies that showed dry-cured ham muscle samples with lower water content had lower springiness and cohesiveness values [31,36]. Other authors also reported a decrease in springiness during the manufacturing process of dry-cured foal ‘cecina’ [38] and in BF muscle of Bayonne ham [9], but in contrast to our results, the authors did not observe a decrease in cohesiveness.

Resilience, defined as the ability of a sample to recover its original height [40], changed significantly (*p* < 0.05) in BF and SM during the production process. The final resilience values in both muscles of the smoked dry-cured ham were similar to the initial values at the beginning of the process.

### 3.3. Effects of Processing Time and Muscle Type on Lipolysis and Proteolysis in Smoked Dry-Cured Ham

During the processing of dry-cured ham, the major components of IMF (triglycerides and phospholipids) undergo an intense lipolysis. Free fatty acids (FFAs) are lipolysis products that affect the final organoleptic characteristics of dry-cured ham. FAAs are precursors of several volatile compounds due to their susceptibility to oxidation [16]. Primary products of lipid oxidation, hydroperoxides, are degraded into secondary oxidation products (aldehydes, ketones, esters, and many others) by a series of complex reactions [41]. A moderate degree of lipid oxidation can have a positive effect on the development of the typical flavor of raw ham, while a higher degree of lipid oxidation leads to an impairment of the nutritional, functional, and sensory properties [2,6,16]. Several factors such as pH, a_w_, NaCl, temperature, and location of the muscle can influence the intramuscular lipids oxidation [16]. The results of lipid oxidation by TBARS method in two muscles (BF and SM) are shown in Figure 1A. Lipid oxidation increased significantly (*p* < 0.05) throughout the production process, from 0.13 to 0.45 mg MDA/kg sample in BF and from 0.13 to 0.52 mg MDA/kg sample in SM. TBARS values remained constant (*p* > 0.05) until the ripening phase, when a significant (*p* < 0.05) increase was observed in both muscles. A similar behavior was also observed in the production of Celta ham [8], where the authors observed the increase in malonaldehyde content during post-salting and dry-ripening (~130 and 245 d of processing). We can assume that an increase in malondialdehyde content occurred slightly later due to the antioxidant effect of smoking. In other studies, increased TBARS values were also observed during the initial production phases of Bayonne [9] and Parma [42] hams. In contrast to our results, the two above mentioned studies reported a sharp decrease in TBARS levels in BF and SM towards the end of the process, which authors [9,42] ascribed to the degradation of unstable aldehydes to volatile compounds and the formation of Schiff bases.

SM had significantly (*p* < 0.05) higher TBARS values than BF at the end of processing. This can be attributed to the higher oxygen exposure and higher NaCl content during the initial stages of smoked dry-cured ham production in SM (Table 1), which resulted in a higher increase in malonaldehyde content. NaCl has been found to impede lipolysis and proteolysis but favor lipid oxidation processes [9,43,44,45]. The final TBARS values of smoked dry-cured ham observed in this study are in agreement with those reported for Bayonne [9] and Istrian [23] ham and higher than those reported for Teruel [34] and Parma ham [42]. Statistical analysis (Table 4) showed a significant (*p* < 0.05) effect of M and P on lipid oxidation during processing of smoked dry-cured ham while the effect of interaction (M × P) was not observed.

Proteolysis is recognized as one of the most important processes for texture and flavor development in dry-cured ham [17]. The intensity of proteolysis is evaluated by the proteolysis index. The values of proteolysis index of different dry-cured ham types range from 22% to 36%, depending on raw material characteristics and technological parameters (mainly NaCl content in ham muscles and ripening time) [14,35,46].

The results of proteolysis index for BF and SM muscle in dry-cured ham are shown in Figure 1B. During the production of smoked dry-cured ham, the proteolysis index increased significantly (*p* < 0.05), from 11.99% to 21.74% in BF and from 12.00% to 16.20% in SM. The proteolysis index for BF showed a continuous increase except after the smoking phase. The most significant increase was observed after the ripening phase, which is consistent with the results of a study of BF and SM of Kraški pršut during the 12-month production process [37]. In contrast, SM showed a lower rate of proteolytic activity during the production of smoked dry-cured ham. The proteolytic index in SM increased after the drying phase and remained constant until the end of the process. At the end of the process, BF showed higher (*p* < 0.05) proteolytic index values, in agreement with the results for Kraški pršut after 12 and 16 months of processing [14] and for Bayonne ham [47]. The lower proteolytic activity in SM results from the lower activity of proteases in SM muscle during processing, due to the significantly (*p* < 0.05) higher NaCl concentrations and the lower (*p* < 0.05) water contents (Table 1), especially in the initial stages of production [9].

Statistical analysis (Table 4) showed that proteolytic activity was significantly (*p* < 0.05) affected by M, P and their interaction (M × P) during the processing of smoked dry-cured ham.

### 3.4. Evaluation of Volatile Compounds during the Processing of Smoked Dry-Cured Ham

The aroma of dry-cured ham is very complex and it is the result of several factors including lipolysis and proteolysis as the most important, but also important is the influence of microbial enzymes, spices used during the salting phase, and the introduction of a smoking step during processing. Table 5 shows the evolution of volatile compounds (percentage of the total area) during the production of smoked dry-cured ham. A total of eighty-eight (88) volatile compounds were identified. Thirty-one compounds were found in raw ham, 40 after salting, 58 after smoking, 60 after drying, and 72 after the ripening phase. Aldehydes and phenols were the predominant groups of compounds with the highest number of volatile compounds (20), followed by alcohols (17), ketones (12), aromatic hydrocarbons (8), aliphatic hydrocarbons (4), esters (4), and terpenes (3).

All chemical groups of compounds were statistically influenced by the production phase (*p* < 0.05), while the type of muscle influenced all volatiles except aromatic and aliphatic hydrocarbons. Regarding the influence of the interaction of muscle and production phase, all chemical groups of compounds were influenced except aliphatic hydrocarbons (Table 4).

Aldehydes were the most abundant compounds in smoked dry-cured ham during the production process (Table 5). The results agreed with other studies that reported that aldehydes as the major compounds in Iberian, Serrano, Parma, Bayonne, and Corsican dry-cured hams [12]. Their content decreased after the drying and ripening phase due to the increase of phenols, especially in the SM muscle. This is in agreement with other authors who found that the aldehydes percentage decreased during the final stages of dry-cured ham processing [8,48]. Regarding the influence of muscle type, this difference was notable after the drying and ripening phases, where aldehydes were present in a greater percentage in BF (49.72 and 38.29%) than in SM (22.22 and 18.74%) muscle. Aldehydes are known to be the main contributors to the unique flavor of dry-cured ham due to their low thresholds and abundance [10]. The most abundant linear aldehydes in smoked dry-cured ham were hexanal, nonanal, octanal, heptanal, and pentanal, which are related to the auto-oxidation of unsaturated fatty acids [49]. Among the linear aldehydes, hexanal was the most abundant, contributing in small amounts to the overall aroma of ham with grassy and fresh notes, but in high concentrations leading to an unpleasant rancid aroma [10]. The different behavior in the two muscles during processing was notable after the drying and the ripening phases, where it was present in higher amounts in BF muscle (39.22 and 10.90%). This difference could be related to the different salt contents in the two muscles, especially after ripening phase. The BF muscle had higher contents (17.55 vs. 11.20% of DM for BF and SM muscles, respectively) because of salt diffusion throughout the ham. Furthermore, Marušić Radovčić et al. (2016) [1] found that samples of smoked dry-cured ham at the end of ripening with higher NaCl content had lower content of aldehydes, which is consistent with our results: the higher the NaCl content during the production process, the lower the aldehyde content. These findings are consistent with the results of other studies on dry-cured ham, which found that the oxidation of unsaturated fatty acids to volatile carbonated compounds is favored in the inner part of the ham, where the higher salt content has a prooxidant role [8,50]. Branched and aromatic aldehydes are formed mainly by the degradation of amino acids. 2-Methylbutanal, 3-methylbutanal, benzaldehyde and benzeneacetaldehyde are very important because of their contribution to the overall aroma of dry-cured ham. Both benzaldehyde and benzeneacetaldehyde increase throughout the production process and are characterized by unpleasant bitter aroma and almond notes. 2-Methylbutanal and 3-methylbutanal followed the same trend. 2-Methylbutanal is associated with nutty, cheesy, and salty notes, while 3-methylbutanal is characterized by fruity, acorn, and cheesy notes [10]. The mentioned branched and aromatic aldehydes showed significant differences between muscles after ripening and had higher amounts in BF muscle than in SM (except benzeneacetaldehyde). The unsaturated aldehydes like 2,4-nonadienal, 2E-decanal, tetradecanal, pentadecanal, hexadecanal, and 9-octadecanal were found only after the ripening phase and were present in higher amounts in BF muscle.

After aldehydes, phenols were the predominant group of compounds with the highest number of volatile compounds (20). Phenols are characteristic compounds of smoked dry-cured ham, especially in Croatian Dalmatinski and Drniški pršut types of smoked dry-cured hams, and are responsible for the unique aroma of smoked meat products. Due to their low threshold, they have a significant influence on the final aroma of the product [51]. The amounts of phenolic compounds are shown in Table 5. As can be seen, a significant difference in phenolic compounds was observed after the smoking phase when 8 phenols were detected, after drying, 10, and after the ripening phase, 18. The type of muscle also influenced the phenol content: SM had higher amounts of phenols than BF due to its external location and exposure to smoking. This content increased as the production process progressed, and the highest levels were found in SM after the ripening phase (52.27%). In BF, after the smoking and drying phases, the internal muscle phenolic compounds had a slower diffusion than SM and their content was 1.92 and 3.03%, respectively. After the ripening phase, a higher amount of phenols was detected in BF muscle (23.22%). The results of this study are in agreement with the phenol content at the end of ripening in Dalmatinski pršut [51]. The most abundant phenols were: 2-methylphenol, 4-methylpheno, 2-methoxyphenol, 3,4-dimethylphenol, and 2,6-dimethoxyphenol which had higher content in SM muscle.

Alcohols were the third main chemical group of compounds observed in the production of smoked dry-cured ham. Alcohol content slightly decreased during the production process in BF muscle (19.97% in raw ham and 15.95% at the end of ripening). However, in SM the alcohol content decreased significantly during the processing (21.23% in raw ham and 6.46% after ripening), which correlates with phenol increase, especially in SM muscle. In this regard, BF showed higher levels of total alcohols than SM, which is in agreement with other studies [14,52]. Seventeen alcohols were identified. Alcohols, especially straight-chain aliphatic alcohols, are formed as reaction products of lipid oxidation, while methyl-branched alcohols may originate from the Strecker degradation of amino acids [52]. Alcohols are often described as an important component of meat volatiles due to their low odor threshold, making them crucial contributors to the aroma of dry-cured meat products [53]. 1-Pentanol, 2-furanmethanol, 1-heptanol, 1-octen-3-ol, 3-octen-1-ol, 1-octanol and phenylethyl alcohol were the most abundant after the ripening phase in the production of smoked dry-cured ham and showed higher amounts in BF than in SM. 2-Furanmethanol, is the alcohol found only in smoked dry-cured ham [51] which is consistent with this study; 2-furanmethanol was found after the smoking phase and showed an increase at the end of the ripening phase. A higher amount was found in the SM muscle, which may be due to the external position of the muscle which is constant with other studies [52]. Alcohols with low odor thresholds found in higher amounts in this study (1-octen-3-ol and 1-octanol) have sharp, fatty, rancid, and mushroom like odors [14]. Moreover, these alcohols are found in almost all studies on dry-cured products and remain in high concentrations during the curing process, as reported by other authors [1,53].

In BF, ketones increased significantly after the salting phase and remained constant at the end of the ripening phase. In SM, a different trend was observed: they increased after the smoking phase, remained constant during the drying phase, and then decreased in the final ripening phase. At the ripening phase ketones had similar amounts in both muscles (7.42 and 8.08% in BF and SM, respectively). The results of this study regarding ketone evolution were similar to those reported by for Celta dry-cured ham [8], in which ketones levels dropped in the final phase of production and showed similar amounts in both BF and SM muscles. In raw ham, 2 ketones were identified, 4 after salting, 8 after smoking and drying and 11 after the ripening phase, indicating that a range of different ketones are formed as the production process progresses. Ketones can be formed by lipid autoxidation or microbial metabolism [10]. Ketones contribute to the aroma of dry-cured ham and have fatty aromas which are associated with cooked meats and the sensory note of blue cheese [52]. 2,5-Octanedienone was the most abundant ketone in all stages of production of smoked dry-cured ham. 2-Ketones have a significant influence on the aroma of meat products due to their large quantities [8]. The 2-ketones detected in the present study were: 2-butanone, 2-heptanone, and 2-nonanone. 2-Heptanone and 2-nonanone were among the most abundant 2-ketones and showed the highest levels after the ripening phase. Their amount was higher in BF muscle. Four ketones (3,5-dimethyl-2(5H)-furanone, 3,4-Dimethyl-2-cyclopenten-1-one, 3-Ethyl-2-hydroxy-2-cyclopenten-1-one, and 1-cyclohexyl-1-propanone) were found only after the ripening phase.

Esters decreased significantly (*p* < 0.05) from the raw piece (4.88 and 2.35% in BF and SM, respectively) to the drying phase and remained constant after the ripening phase (0.99 and 0.47% in BF and SM, respectively). BF showed higher amounts of esters compared to SM throughout the manufacturing process, which is consistent with other studies [8]. Esters greatly enhance the dry-cured ham flavor due to their low odor thresholds. Fruity notes have esters formed from short-chain acids, while esters with long-chain acids have a slightly fatty odor [51]. Ester, isohexyl hexanoate was found only after the ripening phase in both muscles.

Aromatic hydrocarbons have a major impact on the overall aroma due to their lower thresholds. They can originate from lipid oxidation, microbial activity, compounds of plant origin present in the diet, or from the smoking phase of production [8]. Eight (8) aromatic hydrocarbons were detected during the processing of smoked dry-cured ham. Only three aromatic hydrocarbons were detected in the raw ham (benzene, methoxy-phenil-oxime, and 1,3,5-trimethylbenzene). After the drying and smoking phases, there was a statistically significant difference in the content for BF and SM. Following a similar pattern to phenols, aromatic hydrocarbons were more abundant in SM than in BF, which may be attributed to the anatomical location of the muscle.

Aliphatic hydrocarbons, due to their high odor threshold, are not considered important for the aroma of dry-cured hams. Only four aliphatic hydrocarbons were identified: 3,3,4-trimethylheptane, hexane, 1-tridecene, and 3-ethyl-2-methyl-1,3-hexadiene and were present in lower amounts throughout the production process. This is in agreement with another study on smoked dry-cured ham [51].

Terpenes originate from the addition of spices, but also from animal feed that also may be a source of the terpene limonene [51]. However, smoked dry-cured ham is produced without spices and only three terpenes (myrcene, limonene, and linalool) were found in small amounts after the ripening phase (0.71 and 1.00% in BF and SM, respectively).

## 4. Conclusions

In this study, the changes in physicochemical parameters, lipolysis, proteolysis, and volatile compounds depending on the type of muscle (BF and SM) during the processing of smoked dry-cured ham, Dalmatinski pršut, were investigated for the first time. Almost all physicochemical parameters were influenced by muscle type, stage of production, and their interactions. SM had lower water, ash, NaCl content, and a_w_, while fat and protein contents were higher after the ripening stage than in BF. BF showed higher L*a*b* values than SM. Hardness, gumminess, and chewiness were significantly higher in SM than BF, while adhesiveness, springiness, and cohesiveness were significantly higher in BF. During the processing of smoked dry-cured ham, Dalmatinski pršut, an increase in the total amount of volatile compounds was observed. Eighty-eight volatile compounds were identified. An increase in the number of volatile compounds was observed during processing: 31 volatile compounds were identified in the raw ham, 40 after salting, 58 after smoking, 60 after drying, and 72 after the ripening phase. Aldehydes and phenols were the predominant groups of compounds, followed by alcohols, ketones, aromatic hydrocarbons, aliphatic hydrocarbons, esters, and terpenes. Muscle type and production phase significantly affected lipid oxidation and the index of proteolysis; TBARS increased faster in SM than in BF, while proteolysis had an opposite effect and was more pronounced in BF. Higher proteolysis in BF than in SM led to an increase in volatile compounds, which was more pronounced in BF than in SM.

Future work will investigate the effects of prolonged ripening (additional six months of ripening) on the biochemical and textural changes, as well as on the volatile profile of BF and SM in relation to the sensory quality of smoked dry-cured ham.

## Figures and Tables

**Figure 1 foods-10-01228-f001:**
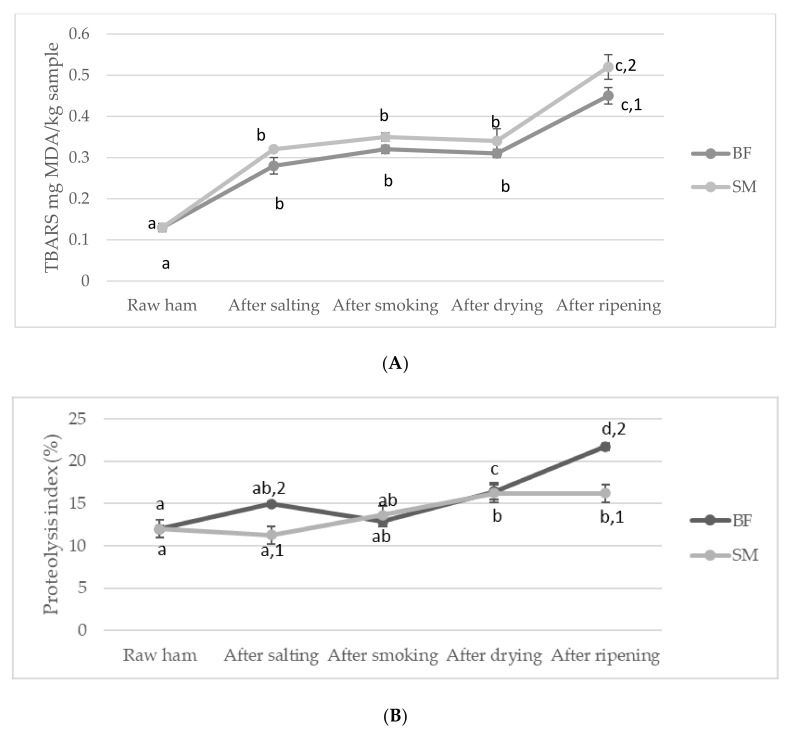
Evolution of lipid oxidation and proteolysis in BF and SM muscles during the production process of smoked dry-cured ham (mean ± standard error): (**A**) Thiobarbituric acid reactive substances (TBARS) index and (**B**) proteolysis index. Different letters (a–d) denote statistically different results (*p* < 0.05; Tukey test) (differences between production stages) while different numbers (1–2) denote statistically different results (*p* < 0.05; Student *t*-test) (differences between muscles).

**Table 1 foods-10-01228-t001:** Physicochemical parameters (expressed as mean ± standard error) in two muscles (biceps femoris (BF) and semimembranosus (SM)) during the manufacturing process of smoked dry-cured ham.

Phase of Production					
	Raw Ham	After Salting	After Smoking	After Drying	After Ripening
Water (%)					
BF	74.82 ± 0.12 ^e^	72.33 ± 0.37 ^d,2^	68.93 ± 0.85 ^c,2^	63.41 ± 0.55 ^b,2^	56.78 ± 0.30 ^a,2^
SM	74.42 ± 0.22 ^d^	64.12 ± 0.36 ^c,1^	63.79 ± 0.29 ^c,1^	59.21 ± 0.53 ^b,1^	42.54 ± 0.75 ^a,1^
**Intramuscular fat (% of dry matter)**					
BF	5.54 ± 0.54 ^b^	5.05 ± 0.19 ^b,1^	2.57 ± 0.18 ^a,1^	5.38 ± 0.44 ^b,1^	12.45 ± 0.45 ^c,1^
SM	6.00 ± 0.44 ^a^	5.72 ± 0.23 ^a,2^	5.55 ± 0.25 ^a,2^	11.02 ± 0.71 ^b,2^	14.86 ± 0.35 ^c,2^
**Protein (% of dry matter)**					
BF	88.19 ± 0.91 ^c^	87.11 ± 1.60 ^c,1^	78.20 ± 0.82 ^b,1^	70.35 ± 1.23 ^a,1^	69.16 ± 1.22 ^a,1^
SM	89.54 ± 1.12 ^c^	75.96 ± 0.89 ^ab,2^	79.82 ± 1.76 ^b,2^	78.26 ± 0.92 ^ab,2^	74.72 ± 0.72 ^a,2^
**Ash (% of dry matter)**					
BF	4.37 ± 0.11 ^a^	8.24 ± 0.35 ^b,1^	13.44 ± 0.32 ^c,1^	17.68 ± 0.35 ^d,1^	19.33 ± 0.48 ^e,2^
SM	4.21 ± 0.08 ^a^	18.97 ± 0.52 ^c,2^	18.21 ± 0.35 ^c,2^	18.36 ± 0.23 ^c,2^	12.91 ± 0.36 ^b,1^
**NaCl (% of dry matter)**					
BF	0.20 ± 0.00 ^a^	4.87 ± 0.44 ^b,1^	12.47 ± 0.20 ^c,1^	15.06 ± 0.32 ^d^	17.55 ± 0.50 ^e,2^
SM	0.18 ± 0.00 ^a^	16.78 ± 0.61 ^c,2^	16.39 ± 0.27 ^c,2^	16.29 ± 0.52 ^c^	11.20 ± 0.35 ^b,1^
**aw**					
BF	0.97 ± 0.00 ^c^	0.97 ± 0.00 ^c,2^	0.96 ± 0.00 ^c,2^	0.93 ± 0.00 ^b,2^	0.88 ± 0.00 ^a,2^
SM	0.96 ± 0.00 ^d^	0.90 ± 0.01 ^b,1^	0.93 ± 0.00 ^c,1^	0.91 ± 0.00 ^bc,1^	0.86 ± 0.00 ^a,1^
**pH**					
BF	5.83 ± 0.05	5.74 ± 0.06	6.02 ± 0.31	5.65 ± 0.02	5.93 ± 0.03
SM	5.78 ± 0.10 ^bc^	5.74 ± 0.04 ^abc^	5.64 ± 0.04 ^ab^	5.51 ± 0.07 ^a^	5.99 ± 0.02 ^c^

^a–e^ Means in the same row (corresponding to the same muscle) not followed by a common letter are significantly different (*p* < 0.05; Tukey test) (differences among sampling points). ^1,2^ Means in the same column and parameter not followed by a common number are significantly different (*p* < 0.05; Student *t*-test) (differences between muscles).

**Table 2 foods-10-01228-t002:** Color parameters (L*a*b*) of smoked dry-cured ham in two muscles (biceps femoris (BF) and semimembranosus (SM)) during the manufacturing process (results expressed as mean ± standard error).

Phase of Production					
	Raw Ham	After Salting	After Smoking	After Drying	After Ripening
L*					
BF	54.02 ± 0.48 ^c^	51.55 ± 0.18 ^a,2^	53.53 ± 0.27 ^bc,2^	54.31 ± 0.47 ^c,2^	52.06 ± 0.40 ^ab,2^
SM	54.41 ± 0.75 ^c^	48.21 ± 0.21 ^ab,1^	49.98 ± 0.57 ^b,1^	49.88 ± 0.41 ^b,1^	47.83 ± 0.34 ^a,1^
a*					
BF	2.96 ± 0.18 ^b,2^	1.95 ± 0.11 ^a^	1.88 ± 0.14 ^a^	1.86 ± 0.16 ^a,2^	2.98 ± 0.18 ^b,2^
SM	1.77 ± 0.09 ^ab,1^	1.84 ± 0.13 ^ab^	1.76 ± 0.14 ^ab^	1.38 ± 0.09 ^a,1^	2.25 ± 0.28 ^b,1^
b*					
BF	8.30 ± 0.37 ^c^	8.32 ± 0.19 ^c^	3.74 ± 0.20 ^a,2^	4.20 ± 0.09 ^a,2^	5.28 ± 0.19 ^b,2^
SM	8.20 ± 0.39 ^c^	8.21 ± 0.14 ^c^	3.26 ± 0.12 ^a,1^	3.41 ± 0.08 ^a,1^	4.47 ± 0.16 ^b,1^

^a–c^ Means in the same row (corresponding to the same muscle) not followed by a common letter are significantly different (*p* < 0.05; Tukey test) (differences among sampling points). ^1,2^ Means in the same column and parameter not followed by a common number are significantly different (*p* < 0.05; Student *t*-test) (differences between muscles). L*: lightness, a*: redness, b*: yellowness.

**Table 3 foods-10-01228-t003:** Textural parameters investigated on two muscles (biceps femoris (BF) and semimembranosus (SM)) through the manufacturing of smoked dry-cured ham (results expressed as mean ± standard error).

Phase of Production					
	Raw Ham	After Salting	After Smoking	After Drying	After Ripening
Hardness (N)					
BF	89.78 ± 3.50 ^b^	93.81 ± 1.71 ^bc^	105.30 ± 5.68 ^c^	92.98 ± 1.74 ^bc,1^	58.00 ± 3.09 ^a,1^
SM	82.21 ± 2.25 ^a^	98.24 ± 1.31 ^b^	114.72 ± 5.15 ^c^	121.04 ± 2.08 ^c,2^	82.62 ± 2.05 ^a,2^
**Gumminess (N)**					
BF	45.50 ± 1.43 ^b^	47.48 ± 1.18 ^b^	51.89 ± 0.49 ^c,1^	43.85 ± 0.52 ^b,1^	23.37 ± 0.99 ^a,1^
SM	42.35 ± 1.24 ^ab^	50.41 ± 3.09 ^bc^	56.51 ± 2.02 ^cd,2^	63.53 ± 0.76 ^d,2^	40.19 ± 2.45 ^a,2^
**Chewiness (N x mm)**					
BF	54.75 ± 1.73 ^b^	61.43 ± 1.37 c^d,1^	69.49 ± 1.18 ^d,1^	59.61 ± 1.29 ^bc,1^	29.87 ± 1.23 ^a,1^
SM	58.21 ± 1.45 ^b^	64.82 ± 0.31 ^c,2^	72.82 ± 0.85 ^d,2^	79.98 ± 1.00 ^e,2^	52.69 ± 0.59 ^a,2^
**Adhesiveness (N x mm)**					
BF	0.94 ± 0.07 ^c^	0.83 ± 0.12 ^bc,2^	0.77 ± 0.04 ^bc,2^	0.59 ± 0.06 ^ab,2^	0.45 ± 0.04 ^a,2^
SM	0.89 ± 0.02 ^d^	0.62 ± 0.04 ^c,1^	0.58 ± 0.02 ^c,1^	0.43 ± 0.01 ^b,1^	0.34 ± 0.02 ^a,1^
**Springiness (mm)**					
BF	0.80 ± 0.08 ^b^	0.85 ± 0.01 ^b,2^	0.91 ± 0.02 ^b,2^	0.56 ± 0.01 ^a,2^	0.45 ± 0.01 ^a,2^
SM	0.75 ± 0.01 ^b^	0.81 ± 0.01 ^c,1^	0.83 ± 0.01 ^c,1^	0.41 ± 0.01 ^a,1^	0.40 ± 0.01 ^a,1^
**Cohesiveness**					
BF	0.50 ± 0.02 ^b,1^	0.49 ± 0.02 ^b^	0.46 ± 0.01 ^b,2^	0.43 ± 0.01 ^ab^	0.38 ± 0.01 ^a,2^
SM	0.59 ± 0.02 ^d,2^	0.53 ± 0.01 ^c^	0.42 ± 0.01 ^b,1^	0.40 ± 0.01 ^b^	0.33 ± 0.01 ^a,1^
**Resilience**					
BF	0.40 ± 0.02 ^a,1^	0.42 ± 0.02 ^a^	0.36 ± 0.02 ^a,1^	0.54 ± 0.03 ^b,2^	0.43 ± 0.01 ^a^
SM	0.47 ± 0.02 ^ab,2^	0.47 ± 0.02 ^ab^	0.50 ± 0.02 ^b,2^	0.47 ± 0.01 ^ab,1^	0.40 ± 0.02 ^a^

^a–d^ Means in the same row (corresponding to the same muscle) not followed by a common letter are significantly different (*p* < 0.05; Tukey test) (differences among sampling points). ^1,2^ Means in the same column and parameter not followed by a common number are significantly different (*p* < 0.05; Student *t*-test) (differences between muscles).

**Table 4 foods-10-01228-t004:** *p*-value for each physicochemical parameter as obtained by GLM analysis.

	Muscle (M)	Phase of Production (P)	M × P
**Physicochemical parameter**			
Water	0.000	0.000	0.000
Intramuscular fat	0.000	0.000	0.000
Protein	0.156	0.000	0.000
Ash	0.000	0.000	0.000
NaCl	0.000	0.000	0.000
a_w_	0.000	0.000	0.000
pH	0.166	0.020	0.329
mg MDA/kg sample	0.003	0.000	0.708
Proteolysis index	0.004	0.000	0.003
**Color parameters**			
L*	0.000	0.000	0.000
a*	0.000	0.000	0.005
b*	0.001	0.000	0.283
**Texture parameters**			
Hardness (N)	0.000	0.000	0.000
Gumminess (N)	0.000	0.000	0.000
Chewiness (N × mm)	0.000	0.000	0.000
Adhesiveness (N × mm)	0.000	0.000	0.000
Springiness (mm)	0.000	0.000	0.301
Cohesiveness	0.965	0.000	0.040
Resilience	0.011	0.000	0.000
**Volatiles**			
Aldehydes	0.000	0.000	0.000
Alcohols	0.000	0.000	0.000
Ketones	0.000	0.000	0.011
Aromatic hydrocarbons	0.047	0.000	0.000
Esters	0.010	0.000	0.000
Aliphatic hydrocarbons	0.539	0.000	0.231
Phenols	0.000	0.000	0.000
Terpenes	0.000	0.000	0.000

*p*-values are obtained by general linear model (GLM) test, at 95% confidence level. Significance levels: *p* > 0.05, not significant; *p* < 0.05, significant.

**Table 5 foods-10-01228-t005:** Evolution of volatile compounds (percentage of the total area) through the manufacturing of smoked dry-cured ham (results expressed as mean ± standard error).

Volatile Compounds	RI	Muscle	Raw Ham	After Salting	After Smoking	After Drying	After Ripening	Identification
Aldehydes								
3-Methylbutanal	645	BF	0.30 ± 0.06 ^b,2^	0.00 ± 0.00 ^a^	0.03 ± 0.00 ^a^	0.07 ± 0.01 ^a^	0.42 ± 0.10 ^b,2^	MS, RI
		SM	0.00 ± 0.00 ^a,1^	0.00 ± 0.00 ^a^	0.04 ± 0.02 ^ab^	0.08 ± 0.02 ^bc^	0.13 ± 0.01 ^c,1^	MS, RI
2-Methylbutanal	654	BF	0.96 ± 0.20 ^b,2^	0.00 ± 0.00 ^a^	0.07 ± 0.01 ^a^	0.14 ± 0.02 ^a^	0.30 ± 0.08 ^a,2^	MS, RI
		SM	0.20 ± 0.12^1^	0.04 ± 0.03	0.05 ± 0.00	0.09 ± 0.05	0.10 ± 0.01^1^	MS, RI
Butanal	657	BF	0.52 ± 0.16 ^b^	0.33 ± 0.08 ^ab^	0.19 ± 0.02 ^ab^	0.30 ± 0.05 ^ab^	0.00 ± 0.00 ^a^	MS, RI
		SM	0.62 ± 0.13 ^c^	0.29 ± 0.05 ^b^	0.16 ± 0.02 ^ab^	0.24 ± 0.02 ^ab^	0.00 ± 0.00 ^a^	MS, RI
Pentanal	713	BF	1.88 ± 0.35 ^ab^	2.57 ± 0.48 ^b^	1.55 ± 0.14 ^ab^	1.49 ± 0.10 ^ab^	0.89 ± 0.16 ^a,2^	MS, RI
		SM	1.23 ± 0.59 ^ab^	1.98 ± 0.22 ^b^	1.08 ± 0.21 ^ab^	2.07 ± 0.47 ^b^	0.17 ± 0.03 ^a,1^	MS, RI
Hexanal	799	BF	44.27 ± 3.18 ^bc^	54.44 ± 2.49 ^c^	44.30 ± 3.06 ^bc^	39.22 ± 2.53 ^b,2^	10.90 ± 0.45 ^a,2^	MS, RI
		SM	48.58 ± 3.16 ^c^	55.53 ± 1.18 ^c^	41.37 ± 1.40 ^b^	7.26 ± 0.67 ^a,1^	1.62 ± 0.07 ^a,1^	MS, RI
Heptanal	903	BF	0.30 ± 0.13 ^a,2^	1.08 ± 0.14 ^ab^	1.87 ± 0.22 ^c^	0.64 ± 0.04 ^a,1^	3.23 ± 0.08 ^d,2^	MS, RI
		SM	0.00 ± 0.00 ^a,1^	1.11 ± 0.14 ^bc^	1.31 ± 0.19 ^c^	1.52 ± 0.27 ^c,2^	0.49 ± 0.06 ^ab,2^	MS, RI
Benzenaldehyde	965	BF	1.70 ± 0.16 ^a^	1.40 ± 0.08 ^a^	3.20 ± 0.84 ^a^	1.08 ± 0.12 ^a,1^	6.50 ± 1.42 ^b,2^	MS, RI
		SM	1.62 ± 0.08 ^a^	1.44 ± 0.11 ^a^	1.51 ± 0.12 ^a^	1.59 ± 0.21 ^a,2^	2.24 ± 0.10 ^b,1^	MS, RI
2,4-Heptadienal	998	BF	0.00 ± 0.00 ^a^	0.00 ± 0.00 ^a,1^	0.26 ± 0.03 ^b,1^	0.17 ± 0.02 ^b,1^	0.51 ± 0.05 ^c,2^	MS, RI
		SM	0.00 ± 0.00 ^a^	0.11 ± 0.03 ^ab,2^	0.68 ± 0.13 ^c,2^	0.37 ± 0.06 ^b,2^	0.00 ± 0.00 ^a,1^	MS, RI
Octanal	1003	BF	0.00 ± 0.00 ^a^	0.00 ± 0.00 ^a^	0.00 ± 0.00 ^a^	1.98 ± 0.31 ^b^	5.24 ± 0.72 ^c,2^	MS, RI
		SM	0.00 ± 0.00 ^a^	0.00 ± 0.00 ^a^	0.00 ± 0.00 ^a^	1.59 ± 0.14 ^b^	1.53 ± 0.11 ^b,1^	MS, RI
Benzeneacetaldehyde	1049	BF	1.04 ± 0.42 ^ab^	0.28 ± 0.07 ^a^	0.28 ± 0.02 ^a^	0.33 ± 0.11 ^a^	1.44 ± 0.15 ^b,1^	MS, RI
		SM	0.84 ± 0.19 ^a^	0.17 ± 0.05 ^a^	0.20 ± 0.04 ^a^	0.35 ± 0.06 ^a^	4.87 ± 1.66 ^b,2^	MS, RI
2-Nonenal	1063	BF	0.21 ± 0.10 ^a^	0.61 ± 0.06 ^c^	0.94 ± 0.07 ^d,1^	0.49 ± 0.06 ^bc,1^	0.33 ± 0.03 ^ab^	MS, RI
		SM	0.16 ± 0.07 ^a^	0.69 ± 0.06 ^ab^	1.63 ± 0.30 ^c,2^	0.88 ± 0.08 ^b,2^	0.31 ± 0.02 ^ab^	MS, RI
Nonanal	1105	BF	3.27 ± 0.40 ^a^	2.71 ± 0.23 ^a^	3.88 ± 0.35 ^ab^	3.40 ± 0.43 ^a.1^	5.31 ± 0.35 ^c,2^	MS, RI
		SM	3.79 ± 0.41 ^a^	2.90 ± 0.29 ^a^	4.01 ± 0.37 ^ab^	5.54 ± 0.67 ^b,2^	3.23 ± 0.14 ^a,1^	MS, RI
Decanal	1207	BF	0.00 ± 0.00 ^a^	0.32 ± 0.05 ^bc,2^	0.27 ± 0.03 ^bc,1^	0.42 ± 0.07 ^c,1^	0.21 ± 0.01 ^b^	MS, RI
		SM	0.00 ± 0.00 ^a^	0.18 ± 0.02 ^a,1^	0.61 ± 0.12 ^b,2^	0.65 ± 0.04 ^b,2^	0.22 ± 0.01 ^a^	MS, RI
2,4-Nonadienal	1213	BF	0.00 ± 0.00 ^a^	0.00 ± 0.00 ^a^	0.00 ± 0.00 ^a^	0.00 ± 0.00 ^a^	0.30 ± 0.03 ^b,2^	MS, RI
		SM	0.00 ± 0.00 ^a^	0.00 ± 0.00 ^a^	0.00 ± 0.00 ^a^	0.00 ± 0.00 ^a^	0.12 ± 0.02 ^b,1^	MS, RI
2E-Decenal	1284	BF	0.00 ± 0.00 ^a^	0.00 ± 0.00 ^a^	0.00 ± 0.00 ^a^	0.00 ± 0.00 ^a^	0.91 ± 0.12 ^b^	MS, RI
		SM	0.00 ± 0.00 ^a^	0.00 ± 0.00 ^a^	0.00 ± 0.00 ^a^	0.00 ± 0.00 ^a^	0.82 ± 0.04 ^b^	MS, RI
2,4-Decadienal	1318	BF	0.00 ± 0.00 ^a^	0.34 ± 0.06 ^c^	0.19 ± 0.02 ^b^	0.00 ± 0.00 ^a^	0.34 ± 0.04 ^c,1^	MS, RI
		SM	0.00 ± 0.00 ^a^	0.24 ± 0.04 ^a^	0.21 ± 0.03 ^a^	0.00 ± 0.00 ^a^	1.94 ± 0.18 ^b,2^	MS, RI
Tetradecanal	1613	BF	0.00 ± 0.00 ^a^	0.00 ± 0.00 ^a^	0.00 ± 0.00 ^a^	0.00 ± 0.00 ^a^	0.26 ± 0.02 ^b,2^	MS, RI
		SM	0.00 ± 0.00 ^a^	0.00 ± 0.00 ^a^	0.00 ± 0.00 ^a^	0.00 ± 0.00 ^a^	0.13 ± 0.03 ^b,1^	MS, RI
Pentadecanal	1716	BF	0.00 ± 0.00 ^a^	0.00 ± 0.00 ^a^	0.00 ± 0.00 ^a^	0.00 ± 0.00 ^a^	0.23 ± 0.02 ^b^	MS, RI
		SM	0.00 ± 0.00 ^a^	0.00 ± 0.00 ^a^	0.00 ± 0.00 ^a^	0.00 ± 0.00 ^a^	0.17 ± 0.03 ^b^	MS, RI
Hexadecanal	1818	BF	0.00 ± 0.00 ^a^	0.00 ± 0.00 ^a^	0.00 ± 0.00 ^a^	0.00 ± 0.00 ^a^	0.73 ± 0.07 ^b,2^	MS, RI
		SM	0.00 ± 0.00 ^a^	0.00 ± 0.00 ^a^	0.00 ± 0.00 ^a^	0.00 ± 0.00 ^a^	0.49 ± 0.08 ^b,1^	MS, RI
9-Octadecanal	1998	BF	0.00 ± 0.00 ^a^	0.00 ± 0.00 ^a^	0.00 ± 0.00 ^a^	0.00 ± 0.00 ^a^	0.24 ± 0.03 ^b^	MS, RI
		SM	0.00 ± 0.00 ^a^	0.00 ± 0.00 ^a^	0.00 ± 0.00 ^a^	0.00 ± 0.00 ^a^	0.17 ± 0.02 ^b^	MS, RI
Total		BF	54.44 ± 3.32 ^bc^	64.09 ± 2.53 ^c^	57.11 ± 2.45 ^bc^	49.72 ± 2.66 ^b,2^	38.29 ± 2.24 ^a,2^	MS, RI
		SM	58.04 ± 3.35 ^bc^	64.68 ± 1.43 ^c^	52.90 ± 1.32 ^b^	22.22 ± 0.89 ^a,1^	18.74 ± 0.58 ^a,1^	MS, RI
Alcohols								
2-Propen-1-ol	625	BF	0.89 ± 0.21 ^b,2^	0.27 ± 0.05 ^a^	0.13 ± 0.02 ^a,1^	1.08 ± 0.23 ^b,2^	0.00 ± 0.00 ^a^	MS, RI
		SM	0.41 ± 0.05 ^bc,1^	0.28 ± 0.06 ^b^	0.24 ± 0.03 ^b,2^	0.48 ± 0.05 ^c,1^	0.00 ± 0.00 ^a^	MS, RI
1-Penten-3-ol	706	BF	0.00 ± 0.00 ^a^	0.24 ± 0.04 ^bc^	0.13 ± 0.03 ^b^	0.25 ± 0.04 ^c,1^	0.16 ± 0.02 ^bc,2^	MS, RI
		SM	0.00 ± 0.00 ^a^	0.20 ± 0.05 ^b^	0.12 ± 0.02 ^b^	0.74 ± 0.00 ^c,2^	0.00 ± 0.00 ^a,1^	MS, RI
3-Buten-1-ol	739	BF	1.65 ± 0.24 ^b,2^	0.10 ± 0.05 ^a,1^	0.10 ± 0.02 ^a^	0.22 ± 0.04 ^a^	0.00 ± 0.00 ^a^	MS, RI
		SM	0.74 ± 0.14 ^c,1^	0.30 ± 0.05 ^b,2^	0.08 ± 0.03 ^ab^	0.16 ± 0.03 ^ab^	0.00 ± 0.00 ^a^	MS, RI
3-Metyl-1-butanol	742	BF	0.32 ± 0.14 ^b^	0.02 ± 0.01 ^a^	0.13 ± 0.01 ^ab,1^	0.35 ± 0.05 ^b^	0.00 ± 0.00 ^a^	MS, RI
		SM	0.09 ± 0.08 ^ab^	0.00 ± 0.00 ^a^	0.19 ± 0.03 ^b,2^	0.40 ± 0.07 ^c^	0.00 ± 0.00 ^a^	MS, RI
1-Pentanol	770	BF	0.00 ± 0.00 ^a,1^	3.40 ± 0.34 ^d,1^	2.76 ± 0.12 ^cd,2^	2.11 ± 0.38 ^bc,2^	1.48 ± 0.34 ^b,2^	MS, RI
		SM	5.34 ± 0.72 ^b,2^	5.44 ± 0.65 ^b,2^	1.71 ± 0.24 ^a,1^	1.17 ± 0.21 ^a,1^	0.54 ± 0.03 ^a,1^	MS, RI
3-Methyl-2-buten-1-ol	778	BF	2.30 ± 0.43 ^b^	0.18 ± 0.06 ^a^	0.24 ± 0.03 ^a^	0.26 ± 0.04 ^a^	0.83 ± 0.15 ^b,2^	MS, RI
		SM	2.15 ± 0.29 ^b^	0.27 ± 0.10 ^a^	0.32 ± 0.05 ^a^	0.20 ± 0.04 ^a^	0.14 ± 0.01 ^a,1^	MS, RI
2-Furanmethanol	859	BF	0.00 ± 0.00 ^a^	0.00 ± 0.00 ^a^	0.24 ± 0.03 ^b,1^	0.50 ± 0.00 ^a,1^	1.21 ± 0.30 ^b,1^	MS, RI
		SM	0.00 ± 0.00 ^a^	0.00 ± 0.00 ^a^	0.54 ± 0.14 ^ab,2^	1.46 ± 0.12 ^bc,2^	2.30 ± 0.63 ^c,2^	MS, RI
1-Hexanol	872	BF	0.00 ± 0.00 ^a^	0.00 ± 0.00 ^a^	0.00 ± 0.00 ^a^	0.98 ± 0.13 ^b,2^	0.00 ± 0.00 ^a,1^	MS, RI
		SM	0.00 ± 0.00 ^a^	0.00 ± 0.00 ^a^	0.00 ± 0.00 ^a^	0.44 ± 0.15 ^b.1^	0.20 ± 0.01 ^ab,1^	MS, RI
1-Heptanol	979	BF	0.91 ± 0.10 ^bc,2^	0.45 ± 0.03 ^a^	0.67 ± 0.04 ^ab,2^	0.49 ± 0.08 ^ab^	1.33 ± 0.21 ^c,2^	MS, RI
		SM	0.40 ± 0.13^1^	0.44 ± 0.04	0.42 ± 0.09^1^	0.44 ± 0.06	0.21 ± 0.03^1^	MS, RI
1-Octen-3-ol	986	BF	4.19 ± 0.36 ^a,1^	8.88 ± 0.81 ^b^	8.73 ± 0.43 ^b,2^	5.19 ± 0.39 ^a,2^	5.31 ± 0.59 ^a,2^	MS, RI
		SM	5.40 ± 0.33 ^b,2^	7.18 ± 0.37 ^c^	6.86 ± 0.35 ^c,1^	2.51 ± 0.21 ^a,1^	1.71 ± 0.15 ^a,1^	MS, RI
2-Ethylhexanol	1035	BF	1.15 ± 0.28 ^b^	0.31 ± 0.03 ^a.2^	0.00 ± 0.00 ^a,1^	1.17 ± 0.06 ^b,2^	0.30 ± 0.07 ^a,2^	MS, RI
		SM	0.72 ± 0.30 ^c^	0.00 ± 0.00 ^a,1^	0.16 ± 0.01 ^ab,2^	0.65 ± 0.07 ^bc,1^	0.14 ± 0.01 ^ab,1^	MS, RI
Benzylalcohol	1037	BF	0.95 ± 0.27 ^b^	0.30 ± 0.07 ^a^	0.30 ± 0.03 ^a^	0.35 ± 0.05 ^a^	0.55 ± 0.03 ^ab,2^	MS, RI
		SM	1.05 ± 0.47 ^b^	0.17 ± 0.02 ^a^	0.46 ± 0.07 ^ab^	0.33 ± 0.03 ^ab^	0.24 ± 0.01 ^ab,1^	MS, RI
3-Octen-1-ol	1073	BF	0.00 ± 0.00 ^a^	0.84 ± 0.10 ^c^	0.92 ± 0.05 ^c,2^	0.45 ± 0.03 ^b,1^	0.74 ± 0.14 ^bc,2^	MS, RI
		SM	0.00 ± 0.00 ^a^	0.71 ± 0.03 ^bc^	0.66 ± 0.09 ^bc,1^	1.03 ± 0.23 ^c,2^	0.41 ± 0.04 ^ab,1^	MS, RI
1-Octanol	1076	BF	7.62 ± 0.82 ^b,2^	1.26 ± 0.10 ^a^	1.56 ± 0.19 ^a,1^	1.35 ± 0.26 ^a^	2.88 ± 0.42 ^a,2^	MS, RI
		SM	4.94 ± 0.26 ^c,1^	1.42 ± 0.11 ^a^	2.03 ± 0.11 ^b,2^	1.42 ± 0.09 ^a^	1.01 ± 0.07 ^a,1^	MS, RI
Phenylethyl alcohol	1113	BF	0.00 ± 0.00 ^a^	0.00 ± 0.00 ^a^	0.12 ± 0.01 ^a^	0.11 ± 0.01 ^a^	0.96 ± 0.12 ^b,2^	MS, RI
		SM	0.00 ± 0.00 ^a^	0.00 ± 0.00 ^a^	0.14 ± 0.02 ^b^	0.11 ± 0.01 ^ab^	0.38 ± 0.06 ^c,1^	MS, RI
1-Undecanol	1174	BF	0.00 ± 0.00 ^a^	0.00 ± 0.00 ^a^	0.09 ± 0.02 ^b,1^	0.07 ± 0.00 ^b,1^	0.00 ± 0.00 ^a^	MS, RI
		SM	0.00 ± 0.00 ^a^	0.00 ± 0.00 ^a^	0.18 ± 0.02 ^b,2^	0.19 ± 0.03 ^b,2^	0.00 ± 0.00 ^a^	MS, RI
2-Cyclohexen-1-ol	1366	BF	0.00 ± 0.00 ^a^	0.00 ± 0.00 ^a^	0.00 ± 0.00 ^a^	0.00 ± 0.00 ^a^	0.51 ± 0.14 ^b^	MS, RI
		SM	0.00 ± 0.00 ^a^	0.00 ± 0.00 ^a^	0.00 ± 0.00 ^a^	0.00 ± 0.00 ^a^	0.34 ± 0.05 ^b^	MS, RI
Total		BF	19.97 ± 1.62 ^b^	16.25 ± 0.98 ^ab^	16.13 ± 0.60 ^ab,2^	14.91 ± 0.53 ^a,2^	15.95 ± 0.90 ^ab,2^	MS, RI
		SM	21.23 ± 1.13 ^d^	16.42 ± 0.71 ^c^	14.12 ± 0.64 ^bc,1^	12.56 ± 0.72^,.1^	6.46 ± 0.10 ^a,1^	MS, RI
Phenols								
2-Methylphenol	1061	BF	0.00 ± 0.00 ^a^	0.00 ± 0.00 ^a^	0.00 ± 0.00 ^a^	0.00 ± 0.00 ^a,1^	2.07 ± 0.36 ^b,1^	MS, RI
		SM	0.00 ± 0.00 ^a^	0.00 ± 0.00 ^a^	0.00 ± 0.00 ^a^	4.58 ± 0.26 ^b,2^	6.05 ± 0.23 ^c,2^	MS, RI
4-Methylphenol	1080	BF	0.00 ± 0.00 ^a^	0.64 ± 0.11 ^a,2^	0.87 ± 0.17 ^a^	0.83 ± 0.09 ^a,1^	5.13 ± 0.54 ^b,1^	MS, RI
		SM	0.00 ± 0.00 ^a^	0.30 ± 0.04 ^ab,1^	1.25 ± 0.15 ^b^	5.87 ± 0.57 ^c,2^	7.22 ± 0.25 ^d,2^	MS, RI
2-Methoxyphenol	1090	BF	0.00 ± 0.00 ^a^	0.00 ± 0.00 ^a^	0.98 ± 0.12 ^ab,1^	2.08 ± 0.09 ^b,1^	9.32 ± 0.85 ^c^	MS, RI
		SM	0.00 ± 0.00 ^a^	0.00 ± 0.00 ^a^	8.28 ± 0.62 ^b,2^	8.26 ± 0.11 ^b,2^	11.08 ± 0.12 ^c^	MS, RI
4-Methoxyphenol	1098	BF	0.00 ± 0.00 ^a^	0.00 ± 0.00 ^a^	0.00 ± 0.00 ^a,1^	0.00 ± 0.00 ^a^	0.38 ± 0.03 ^b^	MS, RI
		SM	0.00 ± 0.00 ^a^	0.00 ± 0.00 ^a^	0.54 ± 0.09 ^b,2^	0.70 ± 0.13 ^b^	0.42 ± 0.02 ^b^	MS, RI
2-Ethylphenol	1140	BF	0.00 ± 0.00 ^a^	0.00 ± 0.00 ^a^	0.00 ± 0.00 ^a^	0.00 ± 0.00 ^a,1^	0.18 ± 0.02 ^b,1^	MS, RI
		SM	0.00 ± 0.00 ^a^	0.00 ± 0.00 ^a^	0.00 ± 0.00 ^a^	0.33 ± 0.07 ^b,2^	0.68 ± 0.04 ^c,2^	MS, RI
2.5-Dimethylphenol (2.5-Xylenol)	1153	BF	0.00 ± 0.00 ^a^	0.00 ± 0.00 ^a^	0.00 ± 0.00 ^a^	0.00 ± 0.00 ^a^	0.33 ± 0.06 ^b,1^	MS, RI
		SM	0.00 ± 0.00 ^a^	0.00 ± 0.00 ^a^	0.00 ± 0.00 ^a^	0.00 ± 0.00 ^a^	2.85 ± 0.20 ^b,2^	MS, RI
2,6-Dimethylphenol	1151	BF	0.00 ± 0.00	0.00 ± 0.00	0.00 ± 0.00^1^	0.00 ± 0.00^1^	0.00 ± 0.00	MS, RI
		SM	0.00 ± 0.00 ^a^	0.00 ± 0.00 ^a^	0.26 ± 0.05 ^b,2^	1.17 ± 0.10 ^c,2^	0.00 ± 0.00 ^a^	MS, RI
3-Ethylphenol	1169	BF	0.00 ± 0.00 ^a^	0.00 ± 0.00 ^a^	0.00 ± 0.00 ^a^	0.00 ± 0.00 ^a,1^	0.43 ± 0.07 ^b,1^	MS, RI
		SM	0.00 ± 0.00 ^a^	0.00 ± 0.00 ^a^	0.00 ± 0.00 ^a^	0.29 ± 0.05 ^b,2^	0.72 ± 0.04 ^c,2^	MS, RI
3,5-Dimethylphenol	1172	BF	0.00 ± 0.00 ^a^	0.00 ± 0.00 ^a^	0.00 ± 0.00 ^a,1^	0.12 ± 0.01 ^b,1^	0.48 ± 0.04 ^c,1^	MS, RI
		SM	0.00 ± 0.00 ^a^	0.00 ± 0.00 ^a^	0.21 ± 0.03 ^a,2^	0.54 ± 0.06 ^b,2^	1.12 ± 0.03 ^c,2^	MS, RI
3,4-Dimethylphenol	1152	BF	0.00 ± 0.00 ^a^	0.00 ± 0.00 ^a^	0.00 ± 0.00 ^a^	0.00 ± 0.00 ^a^	2.00 ± 0.22 ^b,1^	MS, RI
		SM	0.00 ± 0.00 ^a^	0.00 ± 0.00 ^a^	0.00 ± 0.00 ^a^	0.00 ± 0.00 ^a^	9.72 ± 0.47 ^b,2^	MS, RI
2-Methoxy-4-methylphenol	1193	BF	0.00 ± 0.00 ^a^	0.00 ± 0.00 ^a^	0.00 ± 0.00 ^a,1^	0.00 ± 0.00 ^a,1^	0.88 ± 0.25 ^b,1^	MS, RI
		SM	0.00 ± 0.00 ^a^	0.00 ± 0.00 ^a^	1.43 ± 0.49 ^b,2^	0.25 ± 0.03 ^a,2^	2.25 ± 0.16 ^b,2^	MS, RI
3,4-Dimethylphenol	1191	BF	0.00 ± 0.00 ^a^	0.00 ± 0.00 ^a^	0.07 ± 0.01 ^b,1^	0.00 ± 0.00 ^a^	0.00 ± 0.00 ^a^	MS, RI
		SM	0.00 ± 0.00 ^a^	0.00 ± 0.00 ^a^	0.18 ± 0.05 ^b,2^	0.00 ± 0.00 ^a^	0.00 ± 0.00 ^a^	MS, RI
2,3,5-Trimethylphenol	1201	BF	0.00 ± 0.00	0.00 ± 0.00	0.00 ± 0.00	0.00 ± 0.00	0.00 ± 0.00^1^	MS, RI
		SM	0.00 ± 0.00 ^a^	0.00 ± 0.00 ^a^	0.00 ± 0.00 ^a^	0.00 ± 0.00 ^a^	0.35 ± 0.04 ^b,2^	MS, RI
2,4,5-Trimethylphenol	1268	BF	0.00 ± 0.00	0.00 ± 0.00	0.00 ± 0.00	0.00 ± 0.00	0.00 ± 0.00^1^	MS, RI
		SM	0.00 ± 0.00 ^a^	0.00 ± 0.00 ^a^	0.00 ± 0.00 ^a^	0.00 ± 0.00 ^a^	0.40 ± 0.04 ^b,2^	MS, RI
2,4,6-Trimethylphenol	1273	BF	0.00 ± 0.00	0.00 ± 0.00	0.00 ± 0.00	0.00 ± 0.00	0.00 ± 0.00^1^	MS, RI
		SM	0.00 ± 0.00 ^a^	0.00 ± 0.00 ^a^	0.00 ± 0.00 ^a^	0.00 ± 0.00 ^a^	0.50 ± 0.08 ^b,2^	MS, RI
3,4-Dimethoxyphenol	1277	BF	0.00 ± 0.00 ^a^	0.00 ± 0.00 ^a^	0.00 ± 0.00 ^a^	0.00 ± 0.00 ^a^	0.11 ± 0.02 ^b^	MS, RI
		SM	0.00 ± 0.00 ^a^	0.00 ± 0.00 ^a^	0.00 ± 0.00 ^a^	0.00 ± 0.00 ^a^	0.26 ± 0.02 ^b^	MS, RI
4-Ethyl-2-methoxy-phenol	1281	BF	0.00 ± 0.00	0.00 ± 0.00	0.00 ± 0.00^1^	0.00 ± 0.00^1^	0.00 ± 0.00^1^	MS, RI
		SM	0.00 ± 0.00 ^a^	0.00 ± 0.00 ^a^	0.66 ± 0.14 ^b,2^	2.65 ± 0.27 ^c,2^	6.57 ± 0.04 ^d,2^	MS, RI
4-Ethyl-2-methoxyphenol	1291	BF	0.00 ± 0.00 ^a^	0.00 ± 0.00 ^a^	0.00 ± 0.00 ^a^	0.00 ± 0.00 ^a^	0.84 ± 0.12 ^b,2^	MS, RI
		SM	0.00 ± 0.00	0.00 ± 0.00	0.00 ± 0.00	0.00 ± 0.00	0.00 ± 0.00^1^	MS, RI
4-Methoxy-2,3,6-trimethylphenol	1326	BF	0.00 ± 0.00	0.00 ± 0.00	0.00 ± 0.00	0.00 ± 0.00	0.00 ± 0.00^1^	MS, RI
		SM	0.00 ± 0.00 ^a^	0.00 ± 0.00 ^a^	0.00 ± 0.00 ^a^	0.00 ± 0.00 ^a^	0.37 ± 0.08 ^b,2^	MS, RI
2,6-Dimethoxyphenol	1353	BF	0.00 ± 0.00 ^a^	0.00 ± 0.00 ^a^	0.00 ± 0.00 ^a^	0.00 ± 0.00 ^a^	1.09 ± 0.24 ^b,1^	MS, RI
		SM	0.00 ± 0.00 ^a^	0.00 ± 0.00 ^a^	0.00 ± 0.00 ^a^	0.00 ± 0.00 ^a^	1.71 ± 0.11 ^b,2^	MS, RI
Total		BF	0.00 ± 0.00 ^a^	0.64 ± 0.11 ^a,2^	1.92 ± 0.24 ^a,1^	3.03 ± 0.14 ^a,1^	23.22 ± 1.97 ^b,1^	MS, RI
		SM	0.00 ± 0.00 ^a^	0.30 ± 0.04 ^a,1^	12.82 ± 1.43 ^b,2^	24.65 ± 0.91 ^c,2^	52.27 ± 0.37 ^d,2^	MS, RI
Ketones								
2-Butanone	673	BF	0.00 ± 0.00 ^a^	0.00 ± 0.00 ^a^	0.00 ± 0.00 ^a,1^	0.26 ± 0.06 ^b^	0.00 ± 0.00 ^a^	MS, RI
		SM	0.00 ± 0.00 ^a^	0.00 ± 0.00 ^a^	0.87 ± 0.15 ^b,2^	0.28 ± 0.07 ^a^	0.00 ± 0.00 ^a^	MS, RI
2-Heptanone	893	BF	0.00 ± 0.00 ^a^	0.00 ± 0.00 ^a,1^	0.31 ± 0.04 ^a,2^	0.94 ± 0.21 ^b,2^	0.89 ± 0.17 ^b,1^	MS, RI
		SM	0.00 ± 0.00 ^a^	0.14 ± 0.04 ^ab,2^	0.18 ± 0.05 ^b,1^	0.18 ± 0.06 ^b,1^	0.20 ± 0.02 ^b,2^	MS, RI
2-Methyl-2-cyclopenten-1-one	905	BF	0.00 ± 0.00 ^a^	0.00 ± 0.00 ^a^	0.00 ± 0.00 ^a,1^	0.31 ± 0.04 ^b,1^	0.00 ± 0.00 ^a,1^	MS, RI
		SM	0.00 ± 0.00 ^a^	0.00 ± 0.00 ^a^	1.01 ± 0.34 ^b,2^	1.64 ± 0.15 ^b,2^	0.29 ± 0.03 ^a,2^	MS, RI
Octen-3-one	984	BF	0.00 ± 0.00 ^a^	0.00 ± 0.00 ^a^	0.00 ± 0.00 ^a,1^	0.29 ± 0.02 ^b,1^	0.41 ± 0.07 ^b,2^	MS, RI
		SM	0.00 ± 0.00 ^a^	0.00 ± 0.00 ^a^	0.32 ± 0.04 ^a,2^	1.29 ± 0.39 ^b,2^	0.21 ± 0.02 ^a,1^	MS, RI
2,3-Octanedione	990	BF	4.56 ± 0.48 ^b^	6.67 ± 0.62 ^cd^	8.04 ± 0.34 ^d,2^	5.36 ± 0.53 ^bc^	1.27 ± 0.13 ^a,1^	MS, RI
		SM	5.58 ± 0.53 ^abc^	6.89 ± 0.46 ^c^	6.32 ± 0.41 ^bc,1^	4.38 ± 0.86 ^ab^	3.88 ± 0.25 ^a,2^	MS, RI
3,5-Dimethyl-2(5H)-furanone	1001	BF	0.00 ± 0.00 ^a^	0.00 ± 0.00 ^a^	0.00 ± 0.00 ^a^	0.00 ± 0.00 ^a^	0.69 ± 0.09 ^b,2^	MS, RI
		SM	0.00 ± 0.00 ^a^	0.00 ± 0.00 ^a^	0.00 ± 0.00 ^a^	0.00 ± 0.00 ^a^	0.46 ± 0.02 ^b,1^	MS, RI
3,4-Dimethyl-2-cyclopenten-1-one	1022	BF	0.00 ± 0.00 ^a^	0.00 ± 0.00 ^a^	0.00 ± 0.00 ^a^	0.00 ± 0.00 ^a^	0.32 ± 0.04 ^b^	MS, RI
		SM	0.00 ± 0.00 ^a^	0.00 ± 0.00 ^a^	0.00 ± 0.00 ^a^	0.00 ± 0.00 ^a^	0.27 ± 0.02 ^b^	MS, RI
2,3-Dimethyl-2-cyclopenten-1-one	1040	BF	0.00 ± 0.00 ^a^	0.00 ± 0.00 ^a^	0.00 ± 0.00 ^a,1^	0.65 ± 0.07 ^b,1^	1.96 ± 0.17 ^c,2^	MS, RI
		SM	0.00 ± 0.00 ^a^	0.00 ± 0.00 ^a^	2.64 ± 0.44 ^c,2^	2.43 ± 0.38 ^c,2^	1.14 ± 0.07 ^b,1^	MS, RI
3-Octen-2-one	1044	BF	0.00 ± 0.00 ^a^	0.13 ± 0.04 ^b^	0.15 ± 0.01 ^b,1^	0.24 ± 0.04 ^bc,2^	0.32 ± 0.04 ^c^	MS, RI
		SM	0.00 ± 0.00 ^a^	0.09 ± 0.02 ^b^	0.23 ± 0.03 ^cd,2^	0.15 ± 0.02 ^bc,1^	0.27 ± 0.01 ^d^	MS, RI
2-Nonanone	1094	BF	0.22 ± 0.07 ^b^	0.18 ± 0.02 ^ab^	0.09 ± 0.01 ^ab,1^	0.20 ± 0.03 ^ab,1^	1.17 ± 0.08 ^c,2^	MS, RI
		SM	0.24 ± 0.10 ^ab^	0.14 ± 0.03 ^a^	0.27 ± 0.08 ^ab,2^	0.74 ± 0.11 ^bc,2^	0.86 ± 0.04 ^c,1^	MS, RI
3-Ethyl-2-hydroxy-2-cyclopenten-1-one	1111	BF	0.00 ± 0.00 ^a^	0.00 ± 0.00 ^a^	0.00 ± 0.00 ^a^	0.00 ± 0.00 ^a^	0.32 ± 0.03 ^b^	MS, RI
		SM	0.00 ± 0.00 ^a^	0.00 ± 0.00 ^a^	0.00 ± 0.00 ^a^	0.00 ± 0.00 ^a^	0.35 ± 0.02 ^b^	MS, RI
1-Cyclohexyl-1-propanone	1125	BF	0.00 ± 0.00 ^a^	0.00 ± 0.00 ^a^	0.00 ± 0.00 ^a^	0.00 ± 0.00 ^a^	0.14 ± 0.01 ^b,1^	MS, RI
		SM	0.00 ± 0.00 ^a^	0.00 ± 0.00 ^a^	0.00 ± 0.00 ^a^	0.00 ± 0.00 ^a^	0.22 ± 0.01 ^b,2^	MS, RI
Total		BF	4.78 ± 0.47 ^a^	6.98 ± 0.64 ^b^	8.59 ± 0.37 ^b,1^	8.05 ± 0.55 ^b,1^	7.42 ± 0.42 ^b^	MS, RI
		SM	5.82 ± 0.51 ^a^	7.27 ± 0.46 ^a^	11.85 ± 1.22 ^b,2^	11.09 ± 1.04 ^b,2^	8.08 ± 0.13 ^ab^	MS, RI
Esters								
Methyl butanoate	747	BF	1.49 ± 0.21 ^c^	0.32 ± 0.07 ^a,1^	0.38 ± 0.06 ^a,1^	0.94 ± 0.11 ^b,2^	0.00 ± 0.00 ^a^	MS, RI
		SM	1.20 ± 0.18 ^b^	1.02 ± 0.32 ^b,2^	1.01 ± 0.19 ^b,2^	0.24 ± 0.03 ^a,1^	0.00 ± 0.00 ^a^	MS, RI
Ethylhexanoate	1004	BF	3.15 ± 0.81 ^c,2^	1.20 ± 0.07 ^ab^	2.29 ± 0.27 ^bc,2^	0.00 ± 0.00 ^a,1^	0.35 ± 0.05 ^a,2^	MS, RI
		SM	1.16 ± 0.21 ^b,1^	1.38 ± 0.12 ^b^	1.62 ± 0.16 ^b,1^	0.43 ± 0.13 ^a,2^	0.12 ± 0.04 ^a,1^	MS, RI
Isohexyl hexanoate	1345	BF	0.00 ± 0.00 ^a^	0.00 ± 0.00 ^a^	0.00 ± 0.00 ^a^	0.00 ± 0.00 ^a^	0.32 ± 0.12 ^b^	MS, RI
		SM	0.00 ± 0.00 ^a^	0.00 ± 0.00 ^a^	0.00 ± 0.00 ^a^	0.00 ± 0.00 ^a^	0.23 ± 0.02 ^b^	MS, RI
Buthyl benzoate	1346	BF	0.24 ± 0.09 ^ab,2^	0.45 ± 0.06 ^b^	0.39 ± 0.05 ^b,2^	0.00 ± 0.00 ^a^	0.32 ± 0.06 ^b,2^	MS, RI
		SM	0.00 ± 0.00 ^a,1^	0.46 ± 0.08 ^c^	0.22 ± 0.03 ^b,1^	0.00 ± 0.00 ^a^	0.13 ± 0.02 ^ab,1^	MS, RI
Total		BF	4.88 ± 0.80 ^c,2^	1.98 ± 0.09 ^ab,1^	3.06 ± 0.32 ^b^	0.94 ± 0.11 ^a^	0.99 ± 0.18 ^a,2^	MS, RI
		SM	2.35 ± 0.33 ^b,1^	2.87 ± 0.25 ^b,2^	2.86 ± 0.10 ^b^	0.66 ± 0.13 ^a^	0.47 ± 0.01 ^a,1^	MS, RI
Aromatic hydrocarbons								
Benzene	691	BF	0.45 ± 0.28 ^a^	0.14 ± 0.01 ^a^	0.13 ± 0.01 ^a^	2.33 ± 0.12 ^b^	0.28 ± 0.04 ^a,2^	MS, RI
		SM	0.10 ± 0.07 ^a^	0.21 ± 0.04 ^a^	0.10 ± 0.02 ^a^	2.55 ± 0.63 ^b^	0.10 ± 0.00 ^a,1^	MS, RI
1-Methyl-2-propyl-cyclohexane	789	BF	0.00 ± 0.00 ^a^	0.36 ± 0.05 ^b^	0.10 ± 0.02 ^a,1^	0.34 ± 0.08 ^b^	0.00 ± 0.00 ^a^	MS, RI
		SM	0.00 ± 0.00 ^a^	0.33 ± 0.06 ^b^	0.38 ± 0.08 ^b,2^	0.48 ± 0.10 ^b^	0.00 ± 0.00 ^a^	MS, RI
1,4-Dimethyl-benzene (p-xylene)	874	BF	0.00 ± 0.00 ^a^	0.63 ± 0.03 ^b,1^	1.25 ± 0.16 ^c,2^	0.00 ± 0.00 ^a,1^	0.00 ± 0.00 ^a^	MS, RI
		SM	0.00 ± 0.00 ^a^	1.27 ± 0.26 ^b,2^	0.57 ± 0.07 ^ab,1^	0.68 ± 0.31 ^ab,2^	0.00 ± 0.00 ^a^	MS, RI
Methoxy-phenil-oxime	918	BF	6.24 ± 0.81 ^c,2^	0.37 ± 0.05 ^b,2^	0.23 ± 0.09 ^a,1^	0.00 ± 0.00 ^a,1^	0.00 ± 0.00 ^a^	MS, RI
		SM	0.67 ± 0.32^1^	0.00 ± 0.00^1^	0.71 ± 0.18^2^	0.45 ± 0.19^2^	0.00 ± 0.00	MS, RI
2,5-Dimethylfuran	971	BF	0.00 ± 0.00 ^a^	0.00 ± 0.00 ^a^	0.00 ± 0.00 ^a^	0.39 ± 0.04 ^b,1^	0.00 ± 0.00 ^a^	MS, RI
		SM	0.00 ± 0.00 ^a^	0.00 ± 0.00 ^a^	0.00 ± 0.00 ^a^	1.10 ± 0.12 ^b,2^	0.00 ± 0.00 ^a^	MS, RI
1,3,5-Trimethyl-benzene	973	BF	1.07 ± 0.19 ^b^	0.16 ± 0.02 ^a^	0.19 ± 0.02 ^a^	0.82 ± 0.01 ^b,2^	0.36 ± 0.04 ^a,1^	MS, RI
		SM	0.72 ± 0.15 ^b^	0.21 ± 0.04 ^a^	0.32 ± 0.06 ^a^	0.31 ± 0.04 ^a,1^	0.47 ± 0.01 ^ab,2^	MS, RI
1,4-Dimethoxy-benzene	1179	BF	0.00 ± 0.00 ^a^	0.00 ± 0.00 ^a^	0.00 ± 0.00 ^a^	0.00 ± 0.00 ^a,1^	0.34 ± 0.03 ^b,2^	MS, RI
		SM	0.00 ± 0.00 ^a^	0.00 ± 0.00 ^a^	0.00 ± 0.00 ^a^	0.60 ± 0.01 ^b,2^	0.00 ± 0.00 ^a,1^	MS, RI
2,3-Dimetoxytoluene	1275	BF	0.00 ± 0.00 ^a^	0.00 ± 0.00 ^a^	0.00 ± 0.00 ^a^	0.00 ± 0.00 ^a^	0.15 ± 0.03 ^b,1^	MS, RI
		SM	0.00 ± 0.00 ^a^	0.00 ± 0.00 ^a^	0.00 ± 0.00 ^a^	0.00 ± 0.00 ^a^	1.48 ± 0.10 ^b,2^	MS, RI
Total		BF	7.76 ± 0.77 ^c,2^	1.67 ± 0.10 ^a^	1.90 ± 0.15 ^a^	3.89 ± 0.20 ^b,1^	1.13 ± 0.07 ^a,2^	MS, RI
		SM	1.49 ± 0.24 ^a,1^	2.02 ± 0.25 ^a^	2.08 ± 0.19 ^a^	6.16 ± 0.87 ^b,2^	2.04 ± 0.03 ^a,1^	MS, RI
Aliphatic hydrocarbons								
3,3,4-Trimethyl heptane	753	BF	0.53 ± 0.14 ^b^	0.00 ± 0.00 ^a^	0.00 ± 0.00 ^a^	0.00 ± 0.00 ^a,1^	0.00 ± 0.00 ^a^	MS, RI
		SM	0.56 ± 0.26 ^b^	0.00 ± 0.00 ^a^	0.00 ± 0.00 ^a^	0.31 ± 0.06 ^ab,2^	0.00 ± 0.00 ^a^	MS, RI
Hexane	757	BF	0.91 ± 0.26 ^b^	0.00 ± 0.00 ^a^	0.00 ± 0.00 ^a^	0.00 ± 0.00 ^a^	0.00 ± 0.00 ^a^	MS, RI
		SM	0.83 ± 0.34 ^b^	0.00 ± 0.00 ^a^	0.00 ± 0.00 ^a^	0.00 ± 0.00 ^a^	0.00 ± 0.00 ^a^	MS, RI
1-Tridecene	952	BF	0.00 ± 0.00 ^a^	0.12 ± 0.02 ^b,2^	0.11 ± 0.02 ^b^	0.12 ± 0.02 ^b^	0.00 ± 0.00 ^a^	MS, RI
		SM	0.00 ± 0.00 ^a^	0.00 ± 0.00 ^a,1^	0.13 ± 0.02 ^b^	0.14 ± 0.01 ^b^	0.00 ± 0.00 ^a^	MS, RI
3-Ethyl-2-methyl-1,3-hexadiene	1034	BF	0.00 ± 0.00 ^a^	0.00 ± 0.00 ^a^	0.00 ± 0.00 ^a^	0.00 ± 0.00 ^a^	0.64 ± 0.07 ^b,2^	MS, RI
		SM	0.00 ± 0.00 ^a^	0.00 ± 0.00 ^a^	0.00 ± 0.00 ^a^	0.00 ± 0.00 ^a^	0.09 ± 0.01 ^b,1^	MS, RI
Total		BF	1.44 ± 0.37 ^b^	0.12 ± 0.02 ^a,2^	0.11 ± 0.02 ^a^	0.12 ± 0.03 ^a,1^	0.64 ± 0.07 ^a,2^	MS, RI
		SM	1.39 ± 0.45 ^b^	0.00 ± 0.00 ^a,1^	0.13 ± 0.02 ^a^	0.45 ± 0.07 ^a,2^	0.09 ± 0.01 ^a,1^	MS, RI
Terpenes								
Myrcene	994	BF	0.00 ± 0.00 ^a^	1.13 ± 0.09 ^b^	0.97 ± 0.08 ^b^	0.00 ± 0.00 ^a,1^	0.00 ± 0.00 ^a^	MS, RI
		SM	0.00 ± 0.00 ^a^	1.10 ± 0.08 ^b^	1.50 ± 0.28 ^b^	1.10 ± 0.20 ^b,2^	0.00 ± 0.00 ^a^	MS, RI
Limonene	1033	BF	0.00 ± 0.00 ^a^	0.40 ± 0.07 ^b^	0.45 ± 0.05 ^b^	0.28 ± 0.04 ^b,1^	0.71 ± 0.06 ^c,2^	MS, RI
		SM	0.00 ± 0.00 ^a^	0.30 ± 0.07 ^ab^	0.46 ± 0.08 ^b^	0.90 ± 0.16 ^c,2^	0.53 ± 0.05 ^b,1^	MS, RI
Linalool	1101	BF	0.00 ± 0.00 ^a^	0.00 ± 0.00 ^a^	0.12 ± 0.02 ^b,1^	0.12 ± 0.03 ^b,1^	0.00 ± 0.00 ^a,1^	MS, RI
		SM	0.00 ± 0.00 ^a^	0.00 ± 0.00 ^a^	1.00 ± 0.16 ^c,2^	0.54 ± 0.16 ^b,2^	0.47 ± 0.02 ^b,2^	MS, RI
Total		BF	0.00 ± 0.00 ^a^	1.54 ± 0.13 ^c^	1.54 ± 0.15 ^c,1^	0.40 ± 0.06 ^b,1^	0.71 ± 0.06 ^b,1^	MS, RI
		SM	0.00 ± 0.00 ^a^	1.40 ± 0.11 ^b^	2.95 ± 0.40 ^c,2^	2.53 ± 0.29 ^c,2^	1.00 ± 0.06 ^b,2^	MS, RI

^a–d^ Means in the same row (corresponding to the same muscle) not followed by a common letter are significantly different (*p* < 0.05; Tukey test) (differences among sampling points). ^1,2^ Means in the same column and parameter not followed by a common number are significantly different (*p* < 0.05; Student *t*-test) (differences between muscles). RI—retention indices calculated in relation to the retention time of n-alkane (C8–C20); MS—mass spectrum agreed with mass database (NIST 05).

## Data Availability

Not applicable.

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
