# Peer review of "Influence of Muscle Type on Physicochemical Parameters, Lipolysis, Proteolysis, and Volatile Compounds throughout the Processing of Smoked Dry-Cured Ham"

_foods, 2021, doi:10.3390/foods10061228_

Round 1

Reviewer 1 Report

This study was performed to evaluate the influence of two muscle types (Biceps femoris and Semimembranosus) on physicochemical properties and volatile compounds, and the amount of proteolysis and lipolysis during processing. The experimental design is well-fitted with the study objectives. However, the explanation and discussions about the observed results are not well linked.

  1. Major concerns
    1. In this study, because it takes very long periods (12 months) to process the dry-cured ham, microorganism may affect the quality properties of dry-cured ham. Including microbiological analysis are necessary.
    2. The authors should explain the standard of final dry-cured ham such as water contents and aw at the end of processing. In general, what should be the final moisture content and/or aw of dry-cured ham? For example, final water contents of BF and SM were 56% and 42%, respectively. Are they completely dried?
    3. In my opinion, a sensory evaluation should be performed because the authors mentioned that the flavor of dry-cured ham is one of the most important quality attributes that influence consumer acceptance (L60-61). The authors mentioned that IMF content and texture profile affects the appearance, texture, and flavor of dry-cured ham (L181-182). Lipid oxidation, proteolysis and volatile compounds of hams also affected organoleptic properties.
    4. When comparing physicochemical properties of BF and SM, the data must be analyzed using T-test (i.e., water, intramuscular fat, protein etc.).
    5. I have question about the authors’ discussion regarding NaCl content of ham. Because SM had relatively lower water content than BF, NaCl of SM might be higher than that of BF. In dry-cured meat products, the water contents and NaCl concentration cannot be proportional due to osmotic dehydration during pressing period. In addition, the authors need to explain the lower NaCl content of SM after ripening.
    6. L345-347: The authors mentioned SM had higher TBARS values than BF due to greater exposure to oxygen by an external position of SM. It seems to be correlated with IFM contents. Also, NaCl content of BF was higher than that of SM. The authors need to reconsider discussion of lipid oxidation.
    7. The authors should explain why the proteolysis index of SM was lower than that of BF during processing.
    8. Although the content of aldehydes (the most abundant compounds in samples) decreased during the final stages, TBARS values of samples constantly increased. This needs to be explained.
    9. BF had higher NaCl contents than SM at the end of processing (Table 1). The authors commented the higher salt content has a prooxidant role (L411-412). However, TBARs value of SM was higher than that of BF. This needs to be explained more detail.
  2. Minor concerns
    1. L39-41: Please add the references.
    2. The authors should offer the number of replications about all the experiments.
    3. In table 1: Please check the under part of pH and arrange the table.
    4. L186-188: More detail explanation is needed about decreasing protein content.
    5. L228: Table 3 -> Table2.
    6. L230 and L521: L*a*b* -> L*, a* and b*.
    7. L233, why only SM muscle is more exposed to environmental condition?
    8. L243: in SM → in BF
    9. In table 3, the springiness should be arranged.
    10. L275-276, the authors explained that the sharp decrease in hardness was due to the progression of proteolysis. However, SM had the lower proteolysis index compared with that of BF (Fig. 1).
    11. L287: The authors should add more explanation about relation between adhesiveness and salt content. Also, the reference reported low salt ham tend to be less adhesiveness, it is not identical tendency with this study.
    12. L298: Please check this sentence. SM is not higher salt content in this study.
    13. L307: it was not significant difference.
    14. L322-325: Please add the references.
    15. L360-361: Fig.1B shows that proteolysis index of BF was less than 25%, after ripening. However, the authors mentioned the proteolysis index of BF was from 11.99% to 27.94%.
    16. L362: Please check this sentence. Proteolysis is decreased after the smoking phase.
    17. In table 5, RI and MS should be explained using footnote.
    18. The authors should add the footnote in Table 4.
    19. L406-407: The authors should add more explanation about relation between aldehydes and salt contents.
    20. L408: Please check the “DM”. Also, there are many DMs in this manuscript.
    21. L413-415: Please add the references.

Author Response

Response to Reviewer 1

Dear Reviewer 1, thank you for the time and effort you have put into reviewing our work. The comments have encouraged us to carefully revise and refine our manuscript. We hope that the revised manuscript has been improved and is suitable for publication in Journal of Foods.

  1. Major concerns
    1. In this study, because it takes very long periods (12 months) to process the dry-cured ham, microorganism may affect the quality properties of dry-cured ham. Including microbiological analysis are necessary.

Response: That is an interesting comment. Yes, microbiological analysis should be interesting to be done however this was not the aim of our study. Most scientific work on other types of dry-cured ham during its processing include volatile compounds evaluation together with certain physicochemical properties. This research was a part of our project and no microbiological analysis was planned or done. We will certainly consider your comment and keep in mind that in the future project, a more detailed microbiological research can be done on these types of dry-cured hams in the future, but our Laboratory is not working on microbial analysis.

    1. The authors should explain the standard of final dry-cured ham such as water contents and aw at the end of processing. In general, what should be the final moisture content and/or aw of dry-cured ham? For example, final water contents of BF and SM were 56% and 42%, respectively. Are they completely dried?

Response: Thank you for pointing this out. In Specification of smoked dry-cured ham Dalmatinski pršut it is stated that final product at the end of ripening should have water activity bellow 0.93 and water content between 40-60% and that after 12 months of production, the dry-cured ham is ripe and ready for consumption. So, in this study our samples of dry-cured ham had production process of 12 months and were completely dried and ready for the market. This was added in the text in Material and method section L92-95.

    1. In my opinion, a sensory evaluation should be performed because the authors mentioned that the flavor of dry-cured ham is one of the most important quality attributes that influence consumer acceptance (L60-61). The authors mentioned that IMF content and texture profile affects the appearance, texture, and flavor of dry-cured ham (L181-182). Lipid oxidation, proteolysis and volatile compounds of hams also affected organoleptic properties.

Response: Thank you very much for your suggestion. Indeed, flavor of dry-cured ham is important attribute that affects the consumer acceptance and also IMF content and texture analysis was done in this study. However, the aim of this study was to investigate the aroma compounds evaluation together with other important physicochemical characteristics during whole production process of smoked dry-cured ham. We do not believe that conducting a sensory evaluation of raw or "unripened" ham is applicable or appropriate. Our opinion is that a sensory evaluation is beyond the scope of this study. We are currently conducting analyses on smoked dry-cured ham samples after prolonged ripening, and our future work will include analysis of the sensory properties of the BF and SM of smoked dry-cured ham at final stages of production (after ripening and prolonged ripening) that we will show in our future paper. This was added in the text L673-675.

    1. When comparing physicochemical properties of BF and SM, the data must be analyzed using T-test (i.e., water, intramuscular fat, protein etc.).

Response: We sincerely apologize, we made a mistake. It is the case that all measurements have already been analyzed by T-test, but we unintentionally did not mention this in the manuscript. When statistical differences were found between two muscles (p< 0.05), we marked them with different numbers (1, 2) in all tables and figures. We reran the Student t-test and rechecked the data to be sure. The corrections have been made and are now in L 205-206 in the revised manuscript version. We also made a correction in Table 1 for the differences in NaCl contents in BF and SM after drying, as we now did not find significant differences (p>0.05). We also made corrections in the descriptions under all tables.

    1. I have question about the authors’ discussion regarding NaCl content of ham. Because SM had relatively lower water content than BF, NaCl of SM might be higher than that of BF. In dry-cured meat products, the water contents and NaCl concentration cannot be proportional due to osmotic dehydration during pressing period. In addition, the authors need to explain the lower NaCl content of SM after ripening.

Response: Thank you for your comment. Explanation about the lower NaCl content of SM after ripening was added in the text L280-288. (The SM muscle is exposed to a higher concentration of salt during salting and to a higher dehydration during drying and has a higher salt content than the inner BF muscle after the salting, smoking and drying phases. However, it should be noted that throughout the production process there is a tendency to equalize the salt concentration throughout the piece, therefore the salt equalization process results in a higher salt content in BF compared to SM after ripening phase. The diffusion of salt into the muscle tissue from the SM muscle with a higher salt content continues due to the existence of a concentration gradient, which ultimately leads to the inner muscle BF with a higher water content also having a higher salt concentration, as found in this study).

    1. L345-347: The authors mentioned SM had higher TBARS values than BF due to greater exposure to oxygen by an external position of SM. It seems to be correlated with IFM contents. Also, NaCl content of BF was higher than that of SM. The authors need to reconsider discussion of lipid oxidation.

Response: We considered your suggestion and performed the Pearson correlation test for BF and SM with IMF content (%) and TBARS test values. The results were as follows: for BF (r=0.5546, p<0.05) and for SM (r=0.5438, p<0.05), indicating that the correlation is significant (p<0.05) and moderate (0.40-0.69, moderate correlation, *Schober et al., 2018). However, it is not strong, so we cannot conclude that there is a strong correlation between IMF content and lipid oxidation during the production of smoked dry-cured ham and that the higher TBARS levels in SM at the end of processing are a consequence of the higher IMF contents in this muscle. **Bermudez et al. (2014) found no significant correlation between TBARS and IMF content during the production of Celta ham (p > 0.05, r=-0.004).

NaCl content was higher (p<0.05) in BF only at the end of processing. As shown in Table 1, NaCl content in SM was significantly higher (p<0.05) during the initial stages of production, until the drying stage when no differences between muscles were found (p>0.05) and the ripening stage when higher (p<0.05) values were detected in BF. The salt content in BF muscle increases slowly during processing as it is distributed from the outer to the inner part of the ham. This phenomenon occurs due to the absorption of NaCl after salting in SM and subsequently diffuses throughout the ham during processing, resulting in a lower NaCl content in BF during the initial stages of production and a higher content in BF at the end of the process. This effect has been reported in many previous studies, including **Bermudez et al. (2014). In the case of Celta ham (**Bermudez et al., 2014), an equalization of salt content in BF and SM was observed after post-salting (̴ 130 days of processing), similar to our study (after drying, ̴ 165 days). Higher NaCl content was detected in BF after dry-ripening (̴265 days in Celta) and after ripening in our study (̴ 365 days).

As shown in Fig. 1B, SM had higher, but not significantly (p> 0.05) higher TBARS values during processing. The maximum values were reached after the ripening in both muscles. A similar behavior was also observed in the production of Celta ham (**Bermudez et al., 2014), where the authors observed the increase in malonaldehyde content during post-salting and dry-ripening ( ̴ 130 and 245 days of processing). We can hypothesize that an increase in malondialdehyde content occurred slightly later due to the antioxidant effect of smoking.

***Andres et al. (2004) reported that the salt content in BF is not high enough to cause prooxidant effects until long after salting. Thus, we can assume that higher exposure to oxygen and higher NaCl levels during the initial production stages in SM led to an increase in malonaldehyde levels in the second half of the processing of the smoked dry-cured ham.

We have added changes in text in L451-455 and L470-472 in revised version of manuscript. 

*Schober, P., Boer, C., Schwarte, L. A. (2018). Correlation coefficients: appropriate use and interpretation. Anesthesia & Analgesia, 126(5), 1763-1768.

**Bermúdez, R., Franco, D., Carballo, J., Lorenzo, J.M. (2014). Physicochemical changes during manufacture and final sensory characteristics of dry-cured Celta ham. Effect of muscle type. Food Control, 43, 263–269.

***Andrés, A. I., Cava, R., Ventanas, J., Muriel, E., Ruiz, J. (2004). Lipid oxidative changes throughout the ripening of dry-cured Iberian hams with different salt contents and processing conditions. Food Chemistry, 84(3), 375-381.

    1. The authors should explain why the proteolysis index of SM was lower than that of BF during processing.

Response: Thank you for your suggestion. In BF, the higher water content during processing and the lower NaCl content in the initial production stages (Table 1) favor the activity of proteases (*Harkouss et al., 2015), so that BF had a higher proteolysis index compared to SM, which was particularly pronounced after salting and the ripening stage. We have made the corrections in the text (lines 496-499 in the revised manuscript version).

*Harkouss, R., Astruc, T., Lebert, A., Gatellier, P., Loison, O., Safa, H., Portanguen, S., Parafita, E., Mirade, P.S. (2015) Quantitative study of the relationships among proteolysis, lipid oxidation, structure and texture throughout the dry-cured ham process. Food Chem., 166, 522–530.

    1. Although the content of aldehydes (the most abundant compounds in samples) decreased during the final stages, TBARS values of samples constantly increased. This needs to be explained.

Response: Total aldehyde contents (expressed as a percentage of the total area of detected peaks) decreased after the drying and ripening stage due to the increase in phenol content (%) originating from smoke. This is in agreement with *Lorenzo et al. (2013), who found that the percentage of aldehydes decreased during the final stages of dry cured ham processing. Previously, the thiobarbituric acid (TBA) test was thought to be primarily a measure of malondialdehyde (MDA); however, it is now known that TBA reacts with a number of different compounds and is therefore referred to as TBARS (thiobarbituric acid reactive substances). The red pigment, measured at 532 nm, formed by reaction with 2-thiobarbituric acid can be formed by 2-alkenals and 2,4-alkadienals as well as MDA (**Irwin and Hedges, 2004). In the case of the TBARS assay, other substances present in foods such as sugars, acids, esters, amino acids, and oxidised proteins may also react with TBA. Fatty acid composition may also influence the actual level of TBARS that can form in a food sample (**Irwin and Hedges, 2004). Thus, the reason for this result may be the lack of specificity of the TBARS test. We can assume that other substances formed in the muscles of the smoked dry-cured ham as products of intensive proteolysis and lipolysis in the second half of processing reacted with TBA and caused the increase in TBARS values.

*Lorenzo, J.M., Carballo, J., Franco, D. (2013) Effect of the inclusion of chestnut in the finishing diet on volatile compounds of dry-cured ham from Celta pig breed. J. Integr. Agric. 12, 2002–2012.

** Irwin, J.W., Hedges, N. (2004) Measuring lipid oxidation. In: Understanding and Measuring the Shelf-Life of Food (Steele, R., ed.), Woodhead Publishing Series in Food Science, Technology and Nutrition, 289-316.

    1. BF had higher NaCl contents than SM at the end of processing (Table 1). The authors commented the higher salt content has a prooxidant role (L411-412). However, TBARs value of SM was higher than that of BF. This needs to be explained more detail.

Response: Response: Thank you for your comment. NaCl content was significantly higher (p<0.05) in BF only at the end of processing.  As shown in Table 1, NaCl content in SM was significantly higher (p< 0.05) during the initial stages of production, until the drying stage, when no differences were found between muscles (p>0.05), and the ripening stage, when higher (p 0.05) values were found in BF.  This phenomenon occurs due to the distribution of NaCl from the outer to the inner part of the ham (*Bermudez et al, 2014), i.e. from SM to BF. We found a significant increase (p< 0.05) in TBARS values in BF after the ripening phase. However, the salt content in BF is not high enough to cause prooxidant effects until long after salting (**Andres et al, 2004), or at least not greater than those in SM, since NaCl is not the only factor affecting lipid oxidation. SM is positioned external in the ham during production and is thus exposed to oxygen (which increases the lipid oxidation effect). Thus, it can be assumed that the higher exposure to oxygen during the whole processing and the higher NaCl levels during the first production stages in SM led to an increase in malonaldehyde levels in the second half of the processing of the smoked dry-cured ham, resulting in higher TBARS levels at the end of the processing. We have added changes to the text in L471-474 in the revised version of the manuscript.

* Bermúdez, R., Franco, D., Carballo, J., Lorenzo, J.M. (2014). Physicochemical changes during manufacture and final sensory characteristics of dry-cured Celta ham. Effect of muscle type. Food Control, 43, 263–269.

**Andrés, A. I., Cava, R., Ventanas, J., Muriel, E., Ruiz, J. (2004). Lipid oxidative changes throughout the ripening of dry-cured Iberian hams with different salt contents and processing conditions. Food Chemistry, 84(3), 375-381.

  1. Minor concerns
    1. L39-41: Please add the references.

Response: The reference was added.

    1. The authors should offer the number of replications about all the experiments.

Response: Thank you for pointing this out. The number of replications was added in the text L100.

    1. In table 1: Please check the under part of pH and arrange the table.

Response: It was corrected.

    1. L186-188: More detail explanation is needed about decreasing protein content.

Response: Explanation about decreasing protein content was added in the text L247-253. (Also, decrease of the protein content probably occurred due to the absorption of NaCl after salting and subsequent diffusion in muscles during processing, resulting in a lower NaCl content in BF muscle in the initial stages of production and a higher content at the end of the process [8]. A significant (p<0.05) decrease in protein content was observed in both muscles after the salting stage, and it was more pronounced in SM muscle due to its external position and more exposure to environmental conditions.)

    1. L228: Table 3 -> Table2.

Response: It was corrected.

    1. L230 and L521: L*a*b* -> L*, a* and b*.

Response: It was corrected.

    1. L233, why only SM muscle is more exposed to environmental condition?

Response: Explanation why SM is more exposed to environmental conditions was added in the text L307-311. (BF and SM muscles are submitted to different conditions during the dry-cured ham process. BF is covered with the skin and thick layer of fat (internal muscle) and SM muscle is superficial within either skin nor fat cover (external muscle) [8]. Therefore, SM is more exposed to environmental conditions due to its external position.)

    1. L243: in SM → in BF

Response: It was corrected.

    1. In table 3, the springiness should be arranged.

Response: It is arranged

    1. L275-276, the authors explained that the sharp decrease in hardness was due to the progression of proteolysis. However, SM had the lower proteolysis index compared with that of BF (Fig. 1).

Response: Results of analysis of proteolytic index (PI) of BF and SM in smoked dry-cured ham showed that SM had a significantly (p< 0.05) lower PI after salting and ripening stages. An increase in hardness was more pronounced in SM during the smoked dry-cured ham production, especially after salting and drying, which may be attributed to the higher salt concentrations and greater water loss at the initial production stages (Table 1). Hardness is correlated with proteolysis of myofibrillar proteins and higher degree of proteolytic activity leads to tenderisation (decrease in hardness) of muscles, especially at later stages of dry-cured ham processing (*Monin et al., 1997). However, hardness is not only related to PI but is also strongly influenced by water and salt content, and it is known that hardness values increase with water loss and higher salt concentrations (**Ruiz-Ramírez et al., 2006). After ripening, there was a sharp decrease (p<0.05) in hardness in both muscles. Our results showed that this decrease in hardness that occurred between drying and ripening was negligibly higher for SM (38.42 N) than for BF (34.98 N). We can assume that BF did not suffer a greater decrease in hardness at the end of processing due to the incorporation of the salt and the additional loss of water after the ripening stage (it can be seen from Table 1 that the NaCl content in BF significantly (p<0.05) increased after the ripening stage). The incorporation of the salt and the additional water loss in BF likely resulted in higher hardness values that would otherwise decrease due to intense proteolysis in BF muscle (Fig. 1B). We have added the above conclusions to the discussion of the hardness of BF and SM through the smoked dry-cured ham processing (L363-367 in the revised version of the manuscript).

Besides the main factors that affect proteolysis stated above, tenderization of dry-cured ham muscle also depends on the state of proteins (Monin et al., 1997) and their susceptibility to degradation (especially myosin heavy chain, α-actinin, troponin-T, and myosin light chain, which are known to play an important role in the formation of meat texture) (***Zhou et al., 2019). Therefore, our future work will be devoted to the in-depth analysis of the proteolytic changes of the sarcoplasmic and myofibrillar protein fractions during processing in BF and SM with SDS electrophoresis, which will provide new insights to better understand the proteolytic processes and to broaden the knowledge in the field of proteolysis in the production of smoked dry-cured ham.

* Monin, G., Marinova, P., Talmant, A., Martin, J.F., Cornet, M., Lanore, D., Grasso, F. (1997) Chemical and structural changes in dry-cured hams (Bayonne hams) during processing and effects of the dehairing technique. Meat Sci., 47, 29–47.

** Ruiz-Ramírez, J., Arnau, J., Serra, X., Gou, P. (2006) Effect of pH24, NaCl content and proteolysis index on the relationship between water content and texture parameters in biceps femoris and semimembranosus muscles in dry-cured ham. Meat Sci., 72, 185–194.

*** Zhou, C. Y., Wu, J. Q., Tang, C. B., Li, G., Dai, C., Bai, Y., ... & Cao, J. X. (2019). Comparing the proteomic profile of proteins and the sensory characteristics in Jinhua ham with different processing procedures. Food Control, 106, 106694.

    1. L287: The authors should add more explanation about relation between adhesiveness and salt content. Also, the reference reported low salt ham tend to be less adhesiveness, it is not identical tendency with this study.

Response: Thank you for your suggestion. As we indicated in the manuscript, adhesiveness is a parameter strongly related to proteolysis (*López-Pedrouso et al., 2018). *López-Pedrouso et al. (2018) observed that dry-cured ham samples with higher proteolysis indices had increased instrumental adhesiveness and found a significant (p<0.05) positive relationship between proteolysis index and adhesiveness using the Pearson correlation coefficient. In aforementioned study, dry-cured ham samples with higher instrumental adhesiveness had lower, although not significantly lower (p>0.05) NaCl values compared to NaCl values of samples with lower adhesiveness values.

However, since proteolytic processes are affected by salt content and higher salt concentrations decrease protease activity, we can conclude that higher salt concentrations would result in lower adhesiveness values. Since SM is exposed to more intense drying and higher NaCl concentrations in the early stages of production, it is expected that this muscle would be less adhesive. In our study, SM showed significantly (p<0.05) lower adhesiveness values throughout processing, which we attribute to the intense drying and higher salt levels in the early stages of production stages compared to BF. This is also supported by **Andronikov et al. (2013), who concluded that lower salt content and higher water content may have an impact on the increased adhesiveness.

We can conclude that the decrease in adhesiveness in both muscles during smoked dry-cured ham is related to drying and protein gelatinization (***Bozkurt and Bayram, 2006), but BF showed higher (p<0.05) adhesiveness compared to SM due to higher water contents (p<0.05) throughout the processing and possibly higher proteolytic activity at the end of the process.

We have taken your suggestion into account and deleted the reference (9) in the sentence (now in L374), since the relationship between salt and adhesiveness is not directly presented in this work. We have also added the explanation of the differences in adhesiveness values between muscles throughout the smoked dry-cured ham process (L 380-385 in the revised version of the manuscript).

* López-Pedrouso, M., Pérez-Santaescolástica, C., Franco, D., Fulladosa, E., Carballo, J., Zapata, C., Lorenzo, J.M (2018). Comparative proteomic profiling of myofibrillar proteins in dry-cured ham with different proteolysis indices and adhesiveness. Food Chem, 244, 238–245

**Andronikov, D., Gašperlin, L., Polak, T., Žlender, B. (2013). Texture and quality parameters of Slovenian dry-cured ham Kraški pršut according to mass and salt levels. Food Technol. Biotechnol., 51, 112–122. https://hrcak.srce.hr/99755

*** Bozkurt, H., Bayram, M. (2006) Colour and textural attributes of sucuk during ripening. Meat Sci., 73, 344–350

    1. L298: Please check this sentence. SM is not higher salt content in this study.

Response: Thank you for pointing this out. The sentence was corrected.

    1. L307: it was not significant difference.

Response: Thank you for pointing this out. We changed the sentence from: “Resilience increased in both muscles during the first stages of production and returned to values similar to the initial values in BF, while significantly decreased in SM.” To: “The final resilience values in both muscles of the smoked dry-cured ham were similar to the initial values at the beginning of the process.” Changes now stand in L398-400 in the revised version of manuscript.

    1. L322-325: Please add the references.

Response: Thank you for pointing this out, we have added the references that now stand in L445 in the revised version of the manuscript.

    1. L360-361: Fig.1B shows that proteolysis index of BF was less than 25%, after ripening. However, the authors mentioned the proteolysis index of BF was from 11.99% to 27.94%.

Response: Thank you for pointing this out, this was a typographical error. Proteolysis index in BF increased from 11.99% to 21.74%. We have corrected the error in the text in L487 in the revised version of the manuscript.

    1. L362: Please check this sentence. Proteolysis is decreased after the smoking phase.

Response: We apologize, there was a mistake, the proteolysis index for BF showed a continuous increase except after the smoking stage. The corrected sentence is now in L488-489 in the revised version of the manuscript.

    1. In table 5, RI and MS should be explained using footnote.

Response: In table 5 RI and MS are explained using foot note (RI- retention indices calculated in relation to the retention time of n-alkane (C8-C20); MS- mass spectrum agreed with mass database (NIST 05).

    1. The authors should add the footnote in Table 4.

Response: The footnote in Table 4 was added.

    1. L406-407: The authors should add more explanation about relation between aldehydes and salt contents.

Respones: Explanation is added in the text L542-545 (Also Marušić Radovčić et al. (2016) [1] found that samples of smoked dry-cured ham at the end of ripening with higher NaCl content had lower content of aldehydes, which is consistent with our results: the higher the NaCl content during the production process, the lower the aldehyde content).

    1. L408: Please check the “DM”. Also, there are many DMs in this manuscript. ??

Response: When dry matter (DM) was first mention in the text full name and abbreviation was stated. Later in the text only DM was used.

    1. L413-415: Please add the references.

Response: Reference was added.

Reviewer 2 Report

I have the following comments:

  1. The results that were statistically significant should be presented in the abstract.
  2. The smoking processes were not described at all. The studies dealing with smoking of muscles should be included. The following reference should be used: Buchtova, H., Dordevic, D., Duda, I., Honzlova, A., & Kulawik, P. (2019). Modeling Some Possible Handling Ways with Fish Raw Material in Home-Made Sushi Meal Preparation. Foods8(10), 459.
  3. Authors should emphasize more the formation of polycyclic aromatic hydrocarbon (PAH) during smoking processes.
  4. Pricipal component analysis should be included too. Table 2 and 3 should be included in the principal component analysis.
  5. Salt content analysis is not described.
  6. The determination of thiobarbituric acid should be described in detail too, what kind of spectrophotometer was used.
  7. Line 201: significantly p < 0.05?
  8. Line 216: significantly p < 0.05?
  9. Table 4. P values are not explained, when it is significant and how.

Author Response

Response to Reviewer 2

Dear Reviewer 2, thank you for the time and effort you have put into reviewing our work. The comments have encouraged us to carefully revise and refine our manuscript. We hope that the revised manuscript has been improved and is suitable for publication in Journal of Foods.

  1. The results that were statistically significant should be presented in the abstract.

Response: Thank you for your comment. We wanted to put all results that are statistically significant in the abstract, however there is limitation for abstract (200 words maximum) and as there are a lot of results that are statistically significant we cannot put all statistically significant results in the abstract so we hope that present abstract that will be adequate.

  1. The smoking processes were not described at all. The studies dealing with smoking of muscles should be included. The following reference should be used: Buchtova, H., Dordevic, D., Duda, I., Honzlova, A., & Kulawik, P. (2019). Modeling Some Possible Handling Ways with Fish Raw Material in Home-Made Sushi Meal Preparation. Foods, 8(10), 459.

Response: Thank you for pointing this out. Smoking of dry-cured ham samples used in this study was done on whole piece of pigs’ leg and not on each muscle separately. Description of the smoking process was added in the manuscript L90-92.

  1. Authors should emphasize more the formation of polycyclic aromatic hydrocarbon (PAH) during smoking processes.

Response: Thank you for noticing that. During smoking process PAH compound are formed. However, PAH compound analysis was not the aim of this study. We wanted to show aroma evaluation during processing of smoked dry-cured ham together with changes in certain physicochemical parameters. PAH compounds were monitored during processing and their values were within the permitted limits. However, this results were not shown because we are preparing another research paper in which we will discuss more about the formation of PAH compounds during smoked ham production.

  1. Pricipal component analysis should be included too. Table 2 and 3 should be included in the principal component analysis.

Response: We initially wanted to do the PCA analysis, however because we had so many results that were significantly different, PCA graph were cluttered so we decided instead to do the General Linear Model (GLM) to estimate the effects of the factors (muscle (M); phase of production (P) and M x P) on the studied parameters which showed better presentation of the results. For that reason, PCA analysis was not performed.

  1. Salt content analysis is not described.

Response: As explained in the manuscript (Section 2.2 Physicochemical analysis, L101 in the revised version of the manuscript), moisture, fat, ash content, and NaCl were determined according to official methods (Association of Official Analytical Chemists [AOAC] (2000): Official methods of analysis (17th ed.). Gaithersburgh, Maryland: Association of Official Analytical Chemists). In the Instructions for Authors is stated: "...well-established methods may be briefly described and appropriately cited." Since the method for NaCl is described there in detail, and is well known and frequently used, we think there is no need to describe this method in detail.

  1. The determination of thiobarbituric acid should be described in detail too, what kind of spectrophotometer was used.

Response: We have made the changes and detailed the TBARS test. We have added the type of spectrophotometer (Specord 50 Plus, AnalytikJena, Jena, Germany) and also described the specifications of the other equipment used. The changes are in L113-122.

  1. Line 201: significantly p < 0.05?

Response: It was corrected.

  1. Line 216: significantly p < 0.05?

Response: It was corrected.

  1. Table 4. P values are not explained, when it is significant and how.

Response: Thank you for pointing this out. We have made the changes and added the description of significance levels (p>0.05, not significant; p<0.05, significant) in description under the Table 4 and in the section 2.6.

Reviewer 3 Report

Dear authors,

I reviewed the manuscript entitled: “Influence of Muscle Type on Physicochemical Parameters, Lipolysis, Proteolysis, and Volatile Compounds Throughout the Processing of Smoked Dry-cured Ham”. There are minor corrections that require being addressed, therefore please find my comments and suggestions.

  • L16: Please define what green stage means;
  • L34-36: Please rephrase and present this as one sentence;
  • L79: Please clarify just the lean muscles were used, or if any bone was included (bone-in hams versus bone-less hams); It would be great to provide some pictures of the final smoked/cured/ripened hams, either as part of the Manuscript or as Supplementary material.
  • L84: What percentage of salt was used?
  • It will be important to mention how was the cold smoking performed, therefore there’s a need to add more information on the material and methods, as no method is referenced;
  • L86: This should be: from 90 to 70%;
  • L89: Is this the regular time for this type of product?
  • L110: Lean samples not lean colour;
  • L118: N x mm? Please amend along the entire manuscript, including the tables;
  • L 148: Tukey honest?
  • In Table 1 there are very low values for NaCl for BF, please explain and clarify;
  • After Table 1, Table 2 should be presented not Table 4;
  • I recommend to exclude Table 4, and the information from it to be presented in the text, in the results section;
  • Table 3: there should be ‘.’ not ‘,’, please amend accordingly;
  • L312, 344: Tukey test not Tuckey, please amend along all the Manuscript;
  • Table 5 is too large, it consists of 6.5 pages of the Manuscript, which is approximately 30% of the entire paper. I would recommend to amend it in order to reduced it, and eventually present the additional results in the Supplementary Material;
  • L516: Is this statement certain? Please clarify.
  • L520: superscript for w (-> aw);
  • Do authors recommend any further work on this topic? Please add if there are additional challenges that should be addressed.
  • Ref 37 doesn’t have the doi link;
  • The addition of a graphical abstract and figures of the hams will definitely bring a meaningful improvement to the paper, therefore I highly recommend it.

Kind regards and best wishes.

Author Response

Response to Reviewer 3

Dear Reviewer 3, thank you very much for your time and effort in reviewing this manuscript. We hope that the revised manuscript has been improved and is suitable for publication in Journal of Foods.

  • L16: Please define what green stage means;

Response: Green stage means raw ham. Therefore, it was corrected in the abstract as: A total of fifty smoked hams were sampled: raw ham, at their green stage, after salting, smoking, drying, and ripening.

  • L34-36: Please rephrase and present this as one sentence;

Response: It was corrected.

  • L79: Please clarify just the lean muscles were used, or if any bone was included (bone-in hams versus bone-less hams); It would be great to provide some pictures of the final smoked/cured/ripened hams, either as part of the Manuscript or as Supplementary material.

Response: Thank you for your comment. The whole ham was used in the production of the smoked dry-cured ham Dalmatinski pršut, therefore it was bone-in hams. The BF and SM muscles were sampled from ten randomly selected hams at different stages of production. Therefore, just lean muscles BF and SM muscles were taken for the analysis. This was all described in paragraph 2.1. Dry-cured ham samples. Regarding pictures, graphical abstract with pictures of the raw ham and ham after ripening was added.

  • L84: What percentage of salt was used?

Response: Hams were dry-salted with excess of coarse sea salt. A heap was formed alternating layers of ham samples and layers of salt. In this way, the samples were totally covered with salt. This was added in the text L88-90.

  • It will be important to mention how was the cold smoking performed, therefore there’s a need to add more information on the material and methods, as no method is referenced;

Response: It was added in the text L90-92.

  • L86: This should be: from 90 to 70%;

Response: This was corrected

  • L89: Is this the regular time for this type of product?

Response: Yes, this is regular time of production for smoked dry-cured ham Dalmatinski pršut. It is also stated in the text that is produced according to the PGI specification and times of each phase of production are given.

  • L110: Lean samples not lean colour;

Response: It was corrected.

  • L118: N x mm? Please amend along the entire manuscript, including the tables;

Response: It was corrected in the entire manuscript

  • L 148: Tukey honest?

Response: It was corrected.

  • In Table 1 there are very low values for NaCl for BF, please explain and clarify;

Response: At the first stages of the production due to internal position of BF, there is lower values of NaCl content in this muscle. This is changed after ripening where BF had higher NaCl content than SM due to the salt diffusion. Explanation was added in the text in L280-288. (The SM muscle is exposed to a higher concentration of salt during salting and to a higher dehydration during drying and has a higher salt content than the inner BF muscle after the salting, smoking and drying phases. However, it should be noted that throughout the production process there is a tendency to equalize the salt concentration throughout the piece, therefore the salt equalization process results in a higher salt content in BF compared to SM after ripening phase. The diffusion of salt into the muscle tissue from the SM muscle with a higher salt content continues due to the existence of a concentration gradient, which ultimately leads to the inner muscle BF with a higher water content also having a higher salt concentration, as found in this study.)

  • After Table 1, Table 2 should be presented not Table 4;

Response: After table 1, table 2 is presented.

  • I recommend to exclude Table 4, and the information from it to be presented in the text, in the results section

Response: We think that is important to present Table 4 as the results of the General Linear Model (GLM) were used to estimate the effects of the factors (muscle (M); phase of production (P) and M x P) on the studied parameters. From this table it could be seen which effect had significant difference and was used for the discussion therefore we left this table in the main text.

  • Table 3: there should be ‘.’ not ‘,’, please amend accordingly;

Response: It was corrected.

  • L312, 344: Tukey test not Tuckey, please amend along all the Manuscript;

Response: It was corrected in the whole Manuscript

  • Table 5 is too large, it consists of 6.5 pages of the Manuscript, which is approximately 30% of the entire paper. I would recommend to amend it in order to reduced it, and eventually present the additional results in the Supplementary Material;

Response: Thank you for pointing this out. I know that it seems too long, however aroma analysis of dry-cured ham in general consists of a large amount number of volatile compounds and when presenting it is necessary to show all volatile compounds identified so I cannot see how can we shorten this results as volatile compound evaluation during production of smoked dry-cured ham is an important aim of this research. When looking at the scientific literature for aroma evaluation for other dry-cured hams, authors also have very long tables which are presented in the main manuscript and not as supplementary material so we think that is important to show this results in the main manuscript.

  • L516: Is this statement certain? Please clarify.

Response: Yes, this statement is certain because in this research, for the first time the changes in physicochemical parameters, lipolysis, proteolysis and volatile compounds depending on the type of muscle (BF and SM) during the processing of smoked dry-cured ham Dalmatinski pršut were investigated. No data on changes in these parameters during processing (from raw ham to final product) for Dalmatinski pršut was published.

  • L520: superscript for w (-> aw);

Response:it was corrected.

  • Do authors recommend any further work on this topic? Please add if there are additional challenges that should be addressed.

Response: Thank you for your suggestion. A natural progression of this work is to study the effect of prolonged ripening (additional 6 months of ripening) on the chemical and physical properties and on the volatile profile of BF or SM in relation to the sensory quality of smoked dry-cured ham. Prolonged ripening is known to have a significant effect on biochemical and textural changes, but with different intensity in BF and SM (*Bermudez et al., 2014; Cilla et al., 2005; Zhao & Zhao, 2007). We have supplemented the Conclusion section with recommendations for further work in L673-675 in the revised version of the manuscript.

*Bermúdez, R., Franco, D., Carballo, J., Lorenzo, J.M. (2014). Physicochemical changes during manufacture and final sensory characteristics of dry-cured Celta ham. Effect of muscle type. Food Control, 43, 263–269.

Cilla, I., Martínez, L., Beltrán, J. A., Roncalés, P. (2005). Factors affecting acceptability of dry-cured ham throughout extended maturation under “bodega” conditions. Meat Sci., 69(4), 789-795.

Zhou, G. H., & Zhao, G. M. (2007). Biochemical changes during processing of traditional Jinhua ham. Meat Sci., 77(1), 114-120.

  • Ref 37 doesn’t have the doi link;

Response: Thank you for your comment, unfortunately this article does not have the doi link. Instead, we have provided the direct link to the article in the reference (https://hrcak.srce.hr/99755) 

  • The addition of a graphical abstract and figures of the hams will definitely bring a meaningful improvement to the paper, therefore I highly recommend it.

Response: Thank you for your comment. Graphical abstract with figures of the hams was added.

Round 2

Reviewer 1 Report

The authors responded well to the reviewer questions and comments.

Reviewer 2 Report

Authors did not apply PCA analysis and also did not improve the abstract of the manuscript.